# Primordial mimicry induces morphological change in *Escherichia coli*

Hui Lu[1], Honoka Aida[2], Masaomi Kurokawa[2], Feng Chen[3], Yang Xia[1], Jian Xu [1], Kai Li[1], Bei-Wen Ying [2✉] & Tetsuya Yomo [1✉]

The morphology of primitive cells has been the subject of extensive research. A spherical form was commonly presumed in prebiotic studies but lacked experimental evidence in living cells. Whether and how the shape of living cells changed are unclear. Here we exposed the rod-shaped bacterium *Escherichia coli* to a resource utilization regime mimicking a primordial environment. Oleate was given as an easy-to-use model prebiotic nutrient, as fatty acid vesicles were likely present on the prebiotic Earth and might have been used as an energy resource. Six evolutionary lineages were generated under glucose-free but oleic acid vesicle (OAV)-rich conditions. Intriguingly, fitness increase was commonly associated with the morphological change from rod to sphere and the decreases in both the size and the area-to-volume ratio of the cell. The changed cell shape was conserved in either OAVs or glucose, regardless of the trade-offs in carbon utilization and protein abundance. Highly differentiated mutations present in the genome revealed two distinct strategies of adaption to OAV-rich conditions, i.e., either directly targeting the cell wall or not. The change in cell morphology of *Escherichia coli* for adapting to fatty acid availability supports the assumption of the primitive spherical form.

[1] Biomedical Synthetic Biology Research Center, School of Life Sciences, East China Normal University, 3663 North Zhongshan Road, Shanghai 200062, PR China. [2] Graduate School of Life and Environmental Sciences, University of Tsukuba, 1-1-1 Tennoudai, Tsukuba, Ibaraki 305-8572, Japan. [3] School of Software Engineering, East China Normal University, 3663 North Zhongshan Road, Shanghai 200062, PR China. ✉email: ying.beiwen.gf@u.tsukuba.ac.jp; tetsuyayomo@gmail.com

Exploring the shape of primitive cells is crucial to understand the origin of life, as it globally restricts physical and chemical features of a cell[1,2]. Studies on the origin of life generally focused on biochemical reactions with molecular building blocks in prebiotic chemistry and the essentiality of genetic information in synthetic biology[3–6]. Possible routes to the origin of life and further development towards the last universal common ancestor (LUCA) have been intensively studied[7–9]. Successful polynucleotide synthesis from single nucleotides[10] and DNA/RNA replication within vesicles[11,12] revealed the mechanisms by which biochemical components work in protocells. The successful construction of synthetic genomes[13–15] and reduced genomes[16–19] explored the minimal genetic requirements of modern cells. These studies provided fruitful insights and valuable models regarding the building blocks and genetic requirements, possibly for primitive cells; however, primitive morphology has been little studied.

The primitive cell shape remains unknown. Certain morphologies, i.e., shapes and/or sizes, are crucial for cellular life, as they provide a closed space for building blocks to work properly and for genetic materials to perform their biological functions[1,20,21]. Considering the simplicity of the building blocks responsible for primitive cellular life, a spherical structure was assumed and has been employed in model protocells for decades in studies on the origin of life. This is why spherical-shaped compartments, e.g., vesicles and droplets[22–24], are commonly used to mimic protocells[25–27]. However, why a primitive cell may have been spherical and whether spheres were energetically or thermodynamically preferred by primitive cells are still open questions. Since a primitive cell is assumed to have had no cell wall, it might have taken on a spherical shape easily, like the roughly spherical protoplast. In addition, the shapes of modern cells, e.g., bacteria, have been studied based on only genome homology[28]. As one of the experimental demonstrations, L-form bacteria showed irregular morphologies due to deficiencies in the cell wall[29,30]. Most modern bacteria have membrane synthesis machinery[31], which must have arisen during evolution to maintain their shape, e.g., the rod shape, in several microbial genera. Accordingly, the primitive cell without any evolved membrane and/or cell wall might have been spherical in shape[32], but experimental evidence on the shape of primitive cells is needed.

To acquire such experimental evidence, experimental evolution with modern living cells in the laboratory, induced by mimicking the resource regime of primordial environments, has been performed as a trial of devolution[33]. First, oleic acid vesicles (OAVs) were added as an easy-to-use model prebiotic nutrient to mimic the resources available in primordial environments. The chemical composition of the primordial environment for the growth of primitive cells remained controversial[34,35]. The first cellular lifeforms were thought to consume the surrounding fatty acid vesicles for their growth when easily biodegradable small molecules were used up in the prebiotic soup[36,37]. Fatty acid vesicles were likely the major components present on the prebiotic Earth[38] and were adopted as the model membranes of protocells[39]. As a representative model of fatty acid vesicles, oleic acid vesicles (OAVs) may be employed as one carbon resource for early life. Second, E. coli is employed as the cell model, as it is the most representative bacterium of the stable rod shape[40,41]. The morphological change of E. coli from rod to filament is reported as a stress response to starvation[42], which suggests that E. coli is able to change its shape when facing maladaptive resource utilization. Additionally, E. coli grows poorly on oleic acid, which is lethal to other model bacteria, e.g., Bacillus subtills[43–46]. This result indicates that the evolutionary adaptation of E. coli to OAVs (as derivatives of oleic acid) is achievable within the experimental timescale. Finally, the experimental evolution of E. coli employed OAVs as the only carbon source, replacing the commonly used glucose. In the present study, we investigated whether and how E. coli adapts in an OAV-rich environment. We also assessed how cell morphology changes during experimental evolution under this condition.

## Results and discussion

### Experimental evolution of E. coli in an OAV-rich environment.
The laboratory E. coli strain MDS42ΔgalK::Ptet-gfp-kan was used as the cell model because the IS-free small genome of MDS42 was beneficial for precise genome resequencing analysis, and chromosomally incorporated gfp (green fluorescent protein) was practical as an indicator for cell detection and population analysis. We evolved six E. coli populations in either glucose- or OAV-supplemented media for approximately 500 generations (Fig. 1A and Supplementary Data 1). The serial transfer was carried out while maintaining the exponential growth phase (Supplementary Fig. 1A), and the daily cultures were subjected to imaging flow cytometry to quantitatively evaluate the cell concentration, cell morphology, and cellular protein abundance (Supplementary Fig. 1B). The six lineages (L31, 32, 9~12), starting from a common originator (Ori), gradually increased in growth rate over the generations in the presence of OAVs (Fig. 1A, upper). The commonly improved fitness of the six populations (L#) demonstrated that E. coli cells were able to utilize OAVs as their carbon source, which was likely an adaptation to the primordial-like environment rich in fatty acid vesicles. In comparison, the parallel experimental evolution in glucose (G31, 32, 9~12) presented higher growth rates than those seen in the OAV groups but similar dynamics as those in the OAV groups (Fig. 1A, bottom). Intriguingly, the magnitude of fitness improvement in the presence of OAVs was equivalent to that in glucose (Fig. 1B, upper), as the fold changes in growth rates of the evolved (Evo) and original (Ori) populations were not significant between the OAV and glucose groups ($p = 0.1$). This demonstrated that the experimental evolution of a modern bacterium in the laboratory, induced by mimicking the carbon availability of a primordial environment, was applicable and comparable to the evolution in a regular environment with sugar as a carbon source.

### Fitness increase associated with changes in cell shape.
Whether the improved growth fitness was associated with morphological changes was analysed. The cell shape was evaluated by the aspect ratio, which represented the length ratio of the major to minor axes of the cell; that is, the closer to 1 the aspect ratio was, the more spherical the cell. A gradual increase in the mean aspect ratio of the cell population (L#) was commonly observed in the lineages evolved in OAVs (Fig. 1C, upper), unlike what occurred in the lineages evolved in glucose (G#) (Fig. 1C, bottom). Although the experimental evolution raised the aspect ratio independent of the carbon source (fold changes of Evo to Ori >1), the magnitude of the change in OAVs was significantly larger ($p = 0.01$) than that in glucose (Fig. 1B, bottom). The cell morphology was further confirmed by single-cell imaging (Fig. 2 and Supplementary Fig. 2). The cells that evolved in glucose (G#) maintained a rod shape (Fig. 2B), similar to that of Ori cells (Fig. 2A). In contrast, those evolved in OAVs were all shorter and thicker than those evolved in Ori, and some of them were nearly spherical, regardless of whether grown in OAVs or glucose (Fig. 2C). These results demonstrate that rod-shaped E. coli, which generally maintains its shape while metabolizing glucose, became closer to spherical once adapted to OAVs.

Only the cell shape was highly coordinated with the fitness increases, although the fluctuation in other morphological

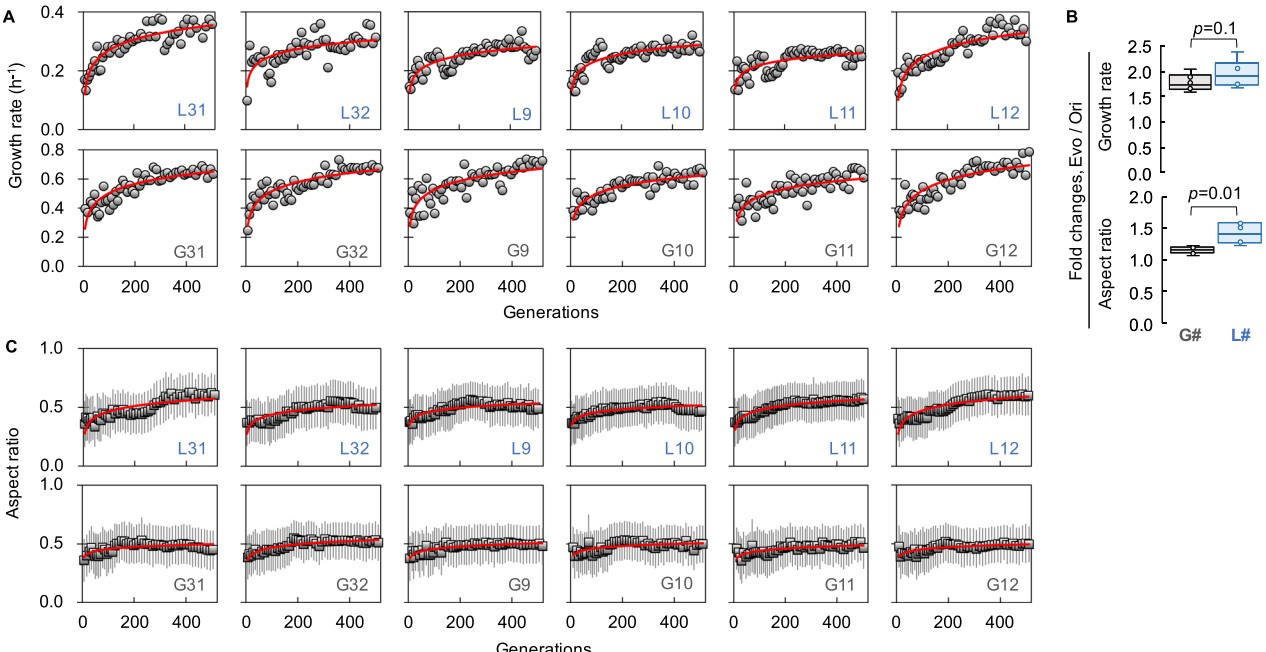

**Fig. 1 Evolutionary changes in the growth rate and morphology of *E. coli*. A** Changes in the growth rate. One out of three cultures of varied dilutions were selected for the following serial transfer. The growth rate of the selected culture was used to assess temporal changes. Logarithmic regression of the temporal changes is represented by the red solid curve. **B** Fold changes in growth and cell shape mediated by experimental evolution. The upper and bottom panels represent the fold changes in growth rate and aspect ratio, respectively. The fold changes were calculated as the ratios between the endpoint populations (L# and G#) and the Ori. Box plots indicate the distributions of the fold changes, in which the six lineages are indicated with open circles. The statistical significance is indicated with the p values. **C** Changes in the cell shape. The cell shape, i.e., the aspect ratio, of the transferred cell populations is shown with respect to those in (**A**). The means and standard deviations were calculated from 10,000 cells subjected to imaging flow cytometry, and they are represented with horizontal and vertical black lines, respectively. Logarithmic regression of the temporal changes is represented by the red solid curve. The labels of the six lineages evolved in OAVs (L#) and glucose (G#) are indicated.

features, i.e., the surface area and volume of the cell, occurred during the evolution (Supplementary Fig. 3). The changes in growth rate were highly correlated with those in aspect ratio and length, which were universally seen in all lineages evolved in OAVs (Fig. 3A and Supplementary Data 2), in contrast to the weak correlation in the lineages evolved in glucose (Fig. 3B and Supplementary Data 3). The fitness increase was tightly associated with the changes in cell shape, which indicated that the consumption of OAVs required the cells to change from rods to spheres. Overall, the laboratory environment mimicking the carbon regime of a primordial environment provides experimental evidence of the sphericity of the cells surrounded by fatty acid vesicles, which supports the speculation of spherical primitive cells on prebiotic Earth.

**Conserved changes in cell shape under supplementation with either OAVs or glucose.** The spherical shape achieved in the adaptation to OAVs seemed to be stable when the cells were grown in the presence of both OAVs and glucose (Fig. 2C), so whether the sphericity could be maintained in the steady-state was further analysed. The aspect ratios of the six Evos, evolved in OAVs, at the steady-state all remained larger than that of the Ori, independent of the concentration of OAVs (0.035–3.5 mM) in the medium (Fig. 4A), indicating conserved changes in cell shape. Changes in the cell shape of these Evos were also observed under glucose-supplemented conditions. The aspect ratio of these Evos remained larger than that of Ori for all concentrations of glucose (0.105–10.5 mM, as OAVs carry threefold more carbon atoms than glucose) (Fig. 4B). The change in cell shape from rods to spheres was conserved, regardless of the carbon source. Although the experimental evolution was performed within the exponential

growth phase, the changed morphology was likely to be fixed independent of the growth phase.

In addition, the *E. coli* cells evolved in OAVs remained shorter and smaller (i.e., area and volume) than Ori cells, irrespective of the concentration of OAVs (Supplementary Fig. 4, upper). Changing to a smaller size in the glucose-free conditions was consistent with the findings that the experimental evolution in glucose-rich conditions had a tendency to adapt *E. coli* cells larger[47]. Highly significant and conserved changes in the cell morphology were identified, even when the evolutionary lineages and OAV abundance were ignored (Fig. 4C–F, upper). However, it was only the changes in cell shape that remained conserved once the *E. coli* cells evolved in OAVs were grown with glucose (Fig. 4C–F, bottom). The cell length and cell size were somehow dependent on the concentration of glucose (Supplementary Fig. 4, bottom). The changes in cell shape toward the spherical form rather than other morphological features were highly crucial for the *E. coli* to use OAVs as the sole carbon source. So far, whether and how OAVs impacted cell morphology through metabolism mediated by molecular machinery is unclear, as both the short-term evolution in OAVs and the long-term evolution in glucose[47,48] cause changes in cell size. Alternatively, the changes in cell shape might be partially attributed to the changes in the plasma membrane capacity of *E. coli* caused by the carbon source changing from glucose to OAVs, because the changes in fatty acid synthesis due to such a nutritional alteration could influence the cell envelope capacity, thereby affecting cell size and morphology[49].

**Trade-offs in carbon utilization efficiency and cellular protein abundance.** An evolutionary trade-off in carbon utilization was

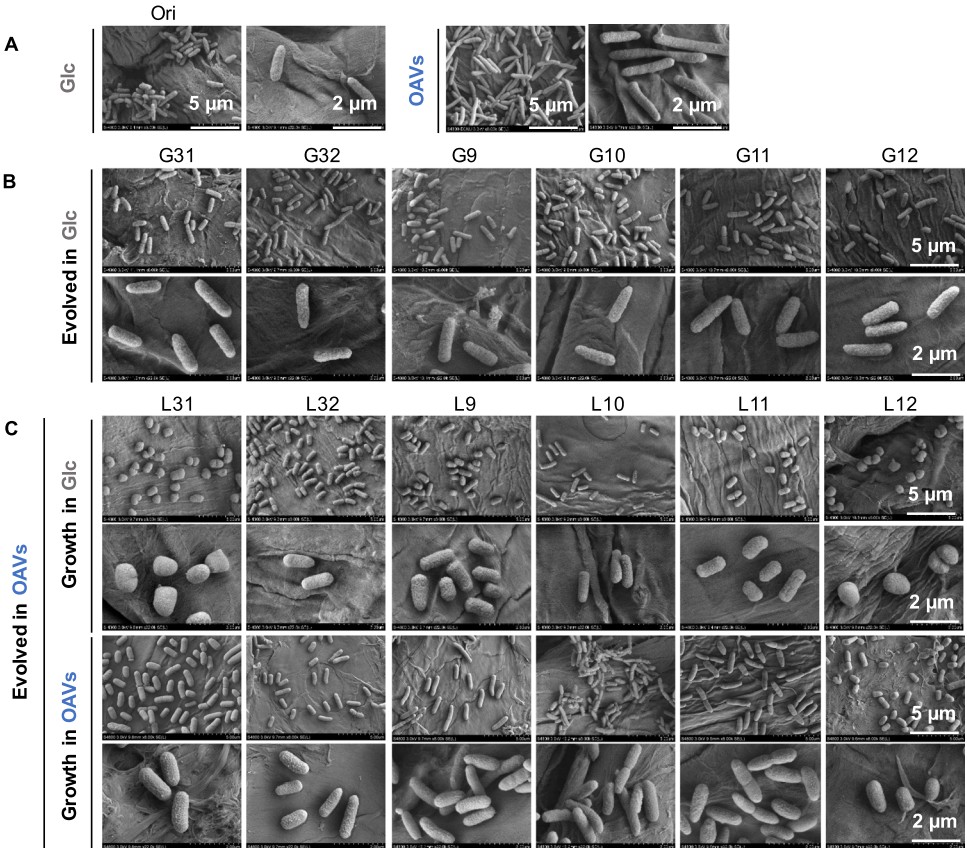

**Fig. 2 Cell shapes imaged by SEM. A** Single-cell images of the Ori in both glucose and OAVs. **B** The six lineages evolved in either glucose or OAVs (**C**) are shown on two size scales. Scale bars are indicated. The upper and bottom panels in (**C**) indicate the lineages evolved in OAVs newly grown in OAVs and glucose, respectively.

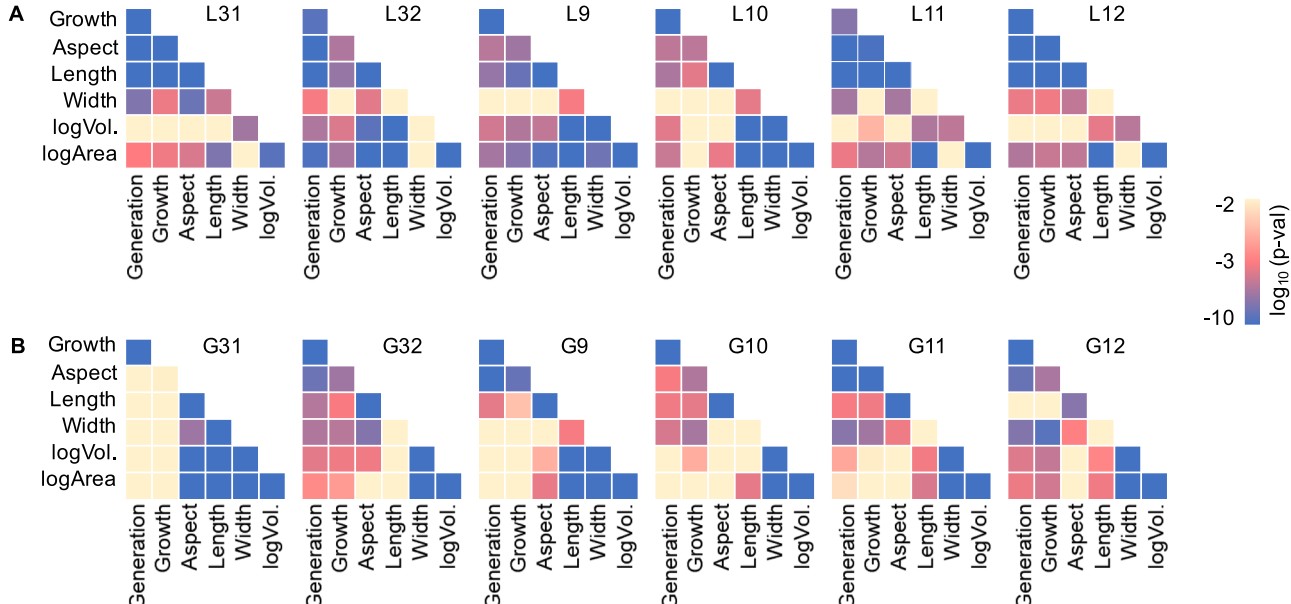

**Fig. 3 Correlations between growth and morphology.** The statistical significance of the correlation coefficients between any two out of six features, which represented the growth and the morphology, is shown in the heatmap. The colour gradient from yellow to blue indicates statistical significance. Blue and purple are highly significant. The six features growth, aspect, length, width, logArea, and logVol. represent the growth rate, aspect ratio, relative cell length (*L*), relative cell width (*W*), and logarithms of the area (*A*) and calculated cell volume (*V*), respectively. The six lineages evolved in either OAVs (**A**) or glucose (**B**) are indicated. The calculated data are summarized in the Supplementary Data 2 and Supplementary Data 3.

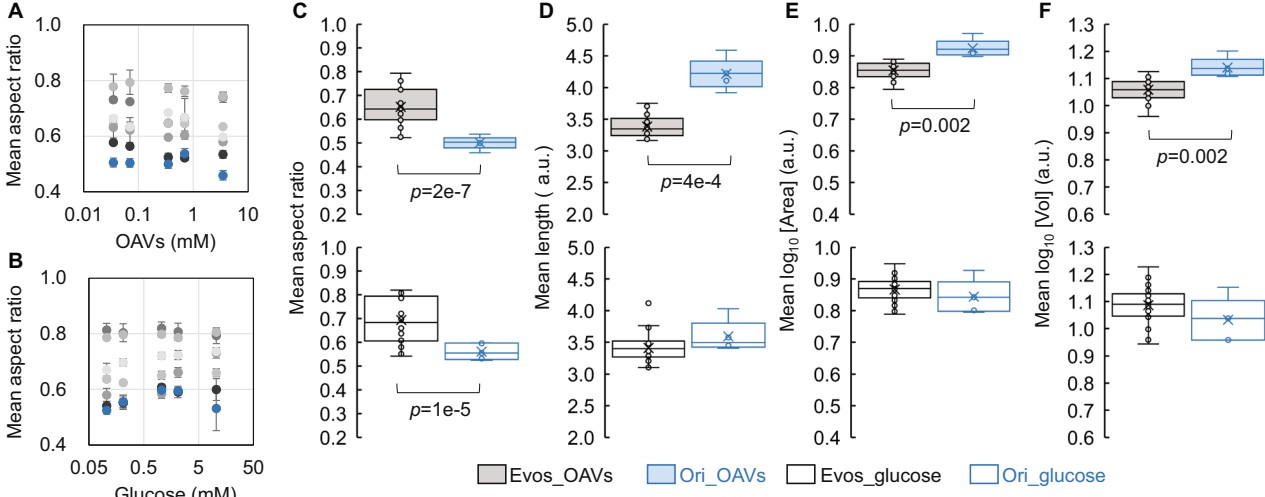

**Fig. 4 Changes in cell shape and cell size.** The cell shape, represented by the mean aspect ratio, in the presence of OAVs (**A**) or glucose (**B**) is shown. The Ori and Evos are indicated as blue and colourless circles, respectively. The grey circles (gradation from light to dark grey) indicate the six lineages. Standard errors of biological repeats ($N > 5$) are indicated. The boxplots of the mean aspect ratio, the mean cell length, the mean logarithmic area, and the volume of the cell populations are shown in (**C**–**F**), respectively. The individual tests are indicated as circles, which correspond to the data in Supplementary Fig. 4. The medians and the averages of the growth rates are represented as lines and crosses inside the box, respectively. Statistical significance is indicated as the $p$-value. The meaning of the colour variation is indicated.

identified, as trade-offs have often occurred in eco-evolution[50,51] and experimental evolution[52,53]. The carbon utilization efficiency was quantitatively represented by the carrying capacity, that is, the maximal cell concentration (Supplementary Fig. 5) per unit carbon source. The carrying capacities of the Evos (evolved in OAVs) showed rough increases with OAVs and decreases with glucose in comparison to those of the Ori (Fig. 5A). Although the trade-offs were dependent on the richness of the carbon source and slightly differed among these Evos, an additional theoretical analysis by means of cubic polynomial regression clearly demonstrated that the carbon utilization efficiency was largely improved for OAVs but decreased or unchanged for glucose (Supplementary Fig. 6).

The trade-off in carrying capacity was linked to the trade-off in cellular protein abundance. The cellular protein abundance was reported by a continuously expressed green florescence protein (GFP) that was chromosomally encoded[54]. The trade-off in cellular protein abundancy occurred in a reverse direction as that in carbon utilization capacity; that is, the GFP concentrations of the Evos were reduced with OAVs but increased with glucose in comparison to those of the Ori (Fig. 5B). The reduced GFP concentrations might be caused by the increased growth rates due to the dilution effect[55]. On the other hand, the GFP concentrations could be increased due to the reduced cell size of the E. coli cells evolved in OAVs. Since not only the relative abundance but also the total amount of GFP were decreased with OAV addition (Supplementary Fig. 7), it was the decreased protein biosynthesis (i.e., transcription and translation) rather than the increased dilution rate caused by the growth increase that caused the reduced protein concentrations when these Evos used OAVs as the carbon source. The fitness gain might be attributed to inexpensive protein biosynthesis for inexpensive carbon catabolism during OAV consumption.

**Decreased area-to-volume ratio caused by changes in cell shape.** The raised aspect ratio was likely caused by the changes in cell length, as the mean length, i.e., the major axis of the cell, became shorter, whereas the width remained the same (Supplementary Fig. 8). It might have been the changes in cell size attributed to the shortened cell length and not the diffusion effect

mediated by the cell width that helped the cells utilize OAVs. In addition to the cell shape (i.e., aspect ratio), the surface area (A) and the volume (V) of the cells were analysed. The area-to-volume (A/V) ratios were significantly decreased with OAV supplementation (Fig. 6A and Supplementary Fig. 9), indicating that the changes in cell shape from rods to spheres reduced the cell area to a greater degree than the cell volume. As the spheres were thought to present a smaller A/V ratio than the rods[56], theoretical predictions suggest that they may impede the diffusion of small molecules from the environment[57,58]. The rod shape with its higher A/V ratio was assumed to be advantageous for utilizing small molecules, e.g., glucose, but not for large molecules, e.g., oleic acids. Alternatively, the small A/V ratio did not inhibit the utilization of fatty acids[58], and the spherical shape was neutral for OAV use. Interestingly, both the Ori and Evos (evolved in OAVs) increased the A/V ratio when facing the replacement of a carbon source, which was supposed to be adapted to glucose and OAVs, respectively. When they were grown under maladaptive conditions, that is, the Ori in OAVs and the Evos in glucose, they all raised the A/V ratios as the first-choice strategy for adaptation. The A/V ratios of the Evos were increased in glucose (Fig. 6A, black) to be comparable to the ratio of the Ori in glucose. Although it was known that the cells became larger with nutrient upshift[49,59], it was unclear whether the glucose-rich condition was nutrient upshift for the Evos, which were evolved in OAVs. The OAVs-adapted E. coli cells probably favour larger A/V ratios in glucose than in OAVs for better diffusion of glucose, to improve glucose utilization. The A/V ratio of the Ori was increased under OAV supplementation (Fig. 6A. blue), although it was unbeneficial. Raising the A/V ratio seemed to be the first choice for adaptation, as most lineages showed increased A/V ratios in the early phase of evolution (Supplementary Fig. 10).

In addition, the slightly but significantly increased A/V ratios failed to improve glucose utilization but did increase the growth rate in glucose irrespective of the glucose concentration or the evolutionary lineage (Fig. 6B and Supplementary Fig. 11). It was highly intriguing that an evolutionary trade-off occurred in carrying capacity but not growth fitness. The adaptation to use OAVs increased the growth fitness for both OAVs and glucose, which contradicted the evolutionary trade-offs in growth

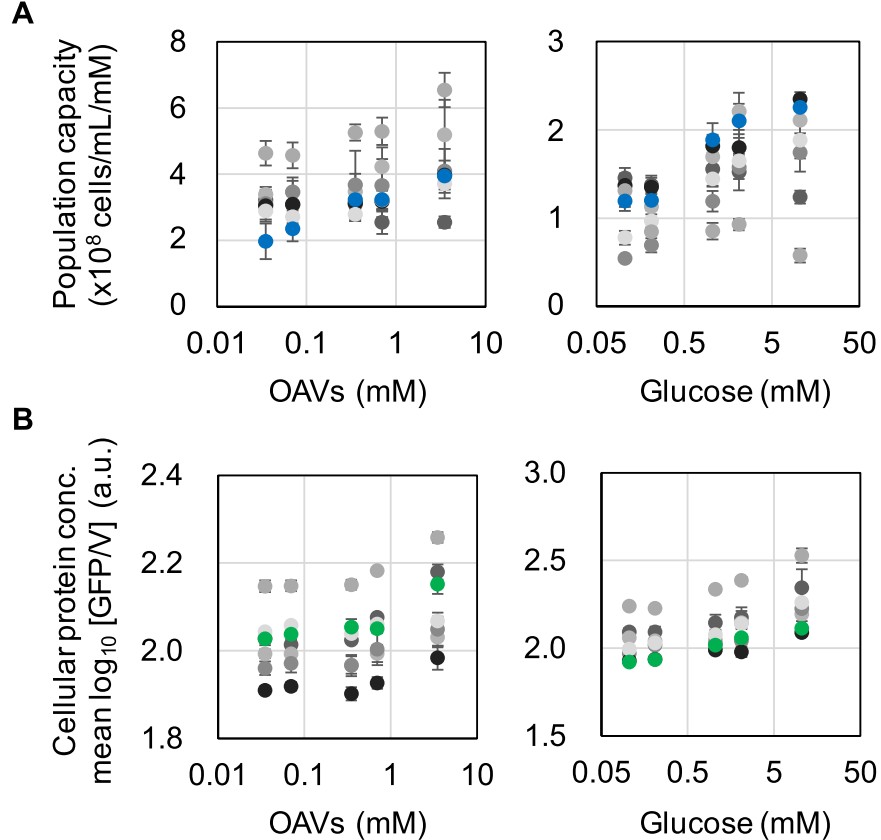

**Fig. 5 Carbon utilization capacity and cellular protein abundance. A** Carrying capacity. The maximal population size (cells/mL) per unit (mM) carbon source (OAVs or glucose) is shown. The Ori and Evos (evolved in OAVs) are indicated as blue and colourless circles, respectively. The grey circles (gradation from light to dark grey) indicate the six lineages. Standard errors of biological replications ($N > 5$) are indicated. **B** Cellular protein abundance. The cellular protein concentration is represented by GFP/V on the logarithmic scale. The Ori and Evos (evolved in OAVs) are indicated as green and colourless circles, respectively. The gradation from light to dark grey indicates the six lineages. Standard errors of biological repeats ($N > 5$) are indicated.

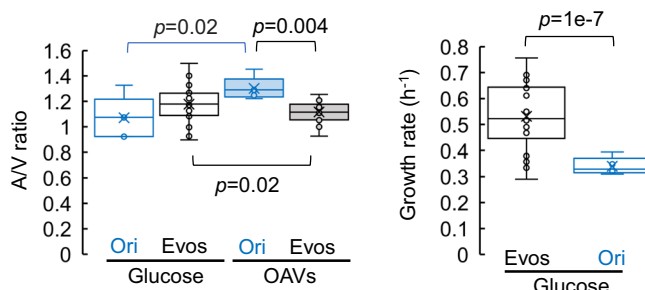

**Fig. 6 Changes in the area-to-volume ratio. A** Boxplots of the area-to-volume ratios. The Ori (blue) and Evos (black) grown in the presence of either OAVs (shadowed boxes) or glucose (open boxes) are shown. The mean area-to-volume (A/V) ratios of the cell populations (measurements) are indicated as circles. The medians and the averages of the mean A/V ratios are represented as lines and crosses inside the box, respectively. Statistical significance is indicated as the *p*-value. **B** Growth rate in glucose. Blue and black indicate the Ori and Evos, respectively. The growth rates of repeated measurements are indicated as circles. The medians and the averages of the growth rates are represented as lines and crosses inside the box, respectively. Statistical significance is indicated as the *p*-value.

fitness[60–62], even when in the same *E. coli* strain[63]. Taken together, the findings suggest that the spherical form might save material and energy for membrane synthesis, which is advantageous in resource-poor conditions or when the cell undergoes

energy-costing metabolism. If the resource of fatty acid vesicles had been deficient on early Earth, the primitive cells might have benefited from the spherical shape by saving resources for growth. The utilization of OAVs might be an energy-costing metabolic pathway for *E. coli* (as devolved pathways in modern cells due to natural evolution); thus, Evos must have favoured spherical shapes to achieve energy-saving growth. If morphological changes are the easiest way to regulate metabolism to achieve efficient growth in response to nutritional changes, it is reasonable that modern cells have evolved molecular machinery for size control, which also affects cell shape as a consequence[49]. Although the nature of both the primitive cell and the primordial environment remains unknown, the present study provides a supportive demonstration of spherical protocells grown in a primordial environment rich in fatty acid vesicles.

**Morphological changes mediated by highly differentiated mutations.** The alteration from rods to spheres of *E. coli* was associated with a wide variety of mutations without common mutations (Fig. 7), suggesting multiple genetic strategies for morphological changes. Intriguingly, the observed mutation was fixed within the same gene, cAMP-activated global transcriptional regulator *crp*[64,65], in three out of the six lineages. As mutations in *crp* have been reported in other evolution experiments with glucose[66–68], transcriptional regulation by *crp* might be crucial for *E. coli* to use carbon sources efficiently. Morphological changes must have been associated with alterations in carbon metabolism to achieve balanced growth fitness. Although *crp* was a shared

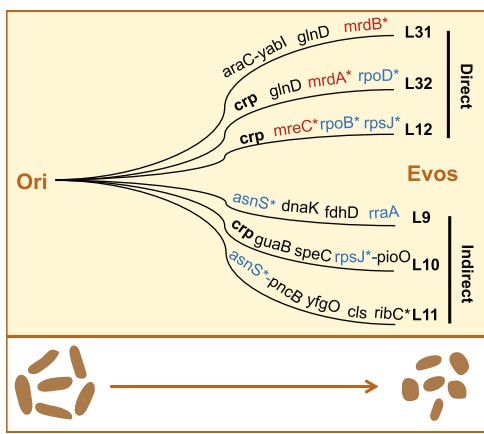

**Fig. 7 Differentiated genetic strategies.** Genome mutations detected in the six Evos are indicated. The essential genes are indicated with asterisks. The genes that played a role in the cellular structure and the cellular protein abundance are highlighted in red and blue, respectively. The mutated gene that appeared in multiple lineages is in bold. The cell shapes of the original and evolved *E. coli* cells are illustrated at the bottom.

mutation target among L32, L10, and L12, the mutated positions varied (Supplementary Data 4 and 5). As the mutations were either nonsynonymous substitutions or deletions (Supplementary Data 4 and 5), disturbing the gene functions, they must benefit the fitness increase-associated changes in cell shape during evolution. Approximately equal numbers of mutations occurred in the six lineages, indicating comparable evolutionary pressure among them.

A common genetic strategy was to target the genes participating in transcription and translation (Fig. 7, blue), which agreed well with the trade-off in cellular protein abundance. Enrichment in mutations in protein biosynthesis supported the hypothesis that countless innovations in translation machinery, genetic codes, etc. occurred during the evolution from primitive to modern life[69]. In addition, diverse strategies for changing the cell shape were noticed. Mutations disturbing cell wall organization occurred in L31, L32, and L12 but not in L9, L10, or L11 (Fig. 7, red). The genes *mrdB*, *mrdA*, and *mreC*, which are responsible for the biosynthesis of peptidoglycan[70,71], the regulation of cell shape[72,73], and the formation of rod shapes[74], respectively, probably directly changed the cell shape from rods to spheres. It was reasonable that the cellular structure-related genes were targeted with high priority during the evolution because they were absent in early life and could be excluded from the minimal genome[13], i.e., they are nonessential for the primitive cell. Note that the direct and indirect strategies could be recognized both in the evolutionary dynamics of growth fitness and cell shape and at the morphological level.

The present study provides the first experimental evidence of rod-shaped bacterial cells changing to spherical shapes in a laboratory environment mimicking the resource regime of the primordial environment, in addition to the intensively reported prebiotic demonstrations of primitive cells. If the rod shape was attributed to the evolved cell wall and other cellular structures, the reverse change from rod to spherical form could be considered a breakdown of well-established cellular structures used by modern cells. The experimental evolution of the fitness-increasing changes in cell shape can be considered morphological devolution to a certain extent. The wide variety of mutations indicates the wide variety of genetic strategies for the evolution of primitive life. Future genetic reconstruction might provide reasonable clarification of the morphological change-associated adaptation to fatty acid vesicles.

Since morphology is regarded as a plastic trait, how long a spherical shape can be maintained in a primordial environment remains unclear. Theoretical prediction according to evolutionary dynamics (Fig. 1) showed that extended evolution would trigger increases in both the growth rate and the aspect ratio, independent of the carbon source (Supplementary Fig. 12). However, the aspect ratio of the cells evolved in OAVs presented high variation among the six lineages, indicating the high plasticity of the spherical shape in the primordial environment. Although the fixed genome mutations indicated the stability of the sphericity, extended evolution in OAVs might lead to new mutations that disturb the morphological features and change the cell shape. The present experimental demonstration with *E. coli* might not represent the universality of morphological determinations of primitive cells; however, it shows some morphological devolution from rods to spheres by a modern bacterium surrounded by fatty acid vesicles. A spherical primitive cell was speculated for decades, probably because of the physical and thermodynamic stability of spheres. The present study provides a reasonable understanding of the growth benefits of spherical cells in primordial environments, and its findings strongly support this speculation, as did prebiotic studies. The ovoid-shaped protocell (LUCA) and the rod-shaped early bacterium (LBCA) were recently reported[75]. Future studies combining synthetic cells and genetic materials are required to fully clarify the original cell shape.

## Materials and methods

***E. coli* strain.** A genetically engineered laboratory *E. coli* strain, *MDS42ΔgalK::P_{tet}-gfp-kan*, which was constructed previously[54], was used for the experimental evolution. The strain carried the reduced genome MDS42, which was originally derived from the wild-type genome MG1655 by removing transposons[16].

**Media**. The OAV-supplemented minimal media were prepared by mixing three stock solutions with ddH$_2$O, resulting in final compositions of 62 mM K$_2$HPO$_4$, 39 mM KH$_2$PO$_4$, 15 mM (NH$_4$)$_2$SO$_4$, 0.009 mM FeSO$_4$, 0.015 mM thiamine hydrochloride, 0.2 mM MgSO$_4$, and 0.035–3.5 mM OAVs. Base, trace element, and OAV solutions were three stock solutions. The base solution, comprising 310 mM K$_2$HPO$_4$, 195 mM KH$_2$PO$_4$, 75 mM (NH$_4$)$_2$SO$_4$, and 0.045 mM FeSO$_4$, was adjusted to pH 8.3 with 2 M KOH. The trace element solution comprised 15 mM thiamine hydrochloride and 203 mM MgSO$_4$. The OAV solution, comprising 5 mM oleic acid vesicles, was prepared according to the method in a previous report[76]. First, the micelle solution was prepared by suspending 316 μL of oleic acid (neat oil) that was degassed in advance in 4.19 mL of alkaline water (pH > 10), which had been prepared by adding 190 μL of 5 M NaOH to 4 mL of ddH$_2$O. The suspended solution (250 mM oleate micelles) was vortexed and agitated overnight at room temperature. Subsequently, the OAV solution was prepared by diluting 250 mM oleate micelles to a final concentration of 5 mM and adjusting to pH 8.3 with HCl (Sinopharm Chemical Reagent Co., Ltd, China). The media were all extruded with an LE-200 extruder (Morgec, China) and a 0.2 μm nucleopore polycarbonate membrane (Whatman, UK) three times and then sterilized with 250 mL filtration units equipped with 0.22 μm membranes (Millipore, USA). Glucose-supplemented minimal media, comprising 62 mM K$_2$HPO$_4$, 39 mM KH$_2$PO$_4$, 15 mM (NH$_4$)$_2$SO$_4$, 0.009 mM FeSO$_4$, 0.015 mM thiamine hydrochloride, 0.2 mM MgSO$_4$ and 0.105–10.5 mM glucose, were prepared as described previously[77]. The media were finally adjusted to pH 8.3 with 2 M KOH, similarly to the OAV-supplemented media. The chemical reagents used for the medium preparation were purchased from Sigma-Aldrich (St. Louis, MO, USA) unless indicated.

**Preparation of the Ori for the experimental evolution**. The *E. coli* strain was initially inoculated into 200 μL of the minimal medium with 3.5 mM OAVs in a flat-bottom 96 microwell plate (Corning, USA) sealed with CyclerSeal Sealing Film (Axygen, USA). The microplates were incubated in a bioshaker (MBR-022UP, Taitec, Japan) with a rotation rate of 200 rpm at 37 °C. Cells passaged several times were subjected to single colony isolation by plating the cell culture on LB agar plates (Supplementary Fig. 1A). Only one of the single colonies was selected as the Ori for the experimental evolution. The selected colony was inoculated in 3 mL of the minimal medium with 3.5 mM OAVs and grown until approximately 10$^8$ cells/mL. The resultant cell culture was preserved as a stock and used as the common Ori of the experimental evolution. Note that a single-nucleotide insertion of T in *ygiM*, encoding the antitoxin *higA*, was initially detected in the Ori in comparison to the *E. coli* strain.

 

**Experimental evolution**. Six independent lineages evolved in either OAVs (L#) or glucose (G#) were generated from the common stock of the Ori described above. The cells were cultured in 3 mL of minimal medium with 3.5 mM OAVs or 10.5 mM glucose, and the cell cultures were incubated in a bioshaker (MBR-022UP, Taitec, Japan) with a rotation rate of 200 rpm at 37 °C. Daily serial transfer to fresh medium was performed at three different dilution rates (Supplementary Fig. 1A), which were estimated according to the daily growth rate as described previously[78]. Only one out of the three cultures in which cell growth was within the exponential phase (e.g., ~$10^7$ cells/mL) was used for the following serial transfer. Note that the initial cell concentration of the daily transfer was higher than $10^4$ cells/mL, to avoid genetic drift. The six lineages were generated independently to avoid cross-contamination. The daily cell cultures were all stocked with 15% glycerol at −80 °C.

**Imaging flow cytometry**. The *E. coli* cell populations were analysed using an Amnis™ ImageStream™X imaging flow cytometer installed with INSPIRE acquisition software (Luminex, USA). Green fluorescence was induced with a 200 mW 488 nm laser, and the emission was detected with a 505–560 nm filter in channel 2. The bright field data were collected in channel 4, side scatter (SSC) was produced with a 2 mW 785 nm laser, and emissions were collected in channel 6 with a 745–800 nm filter. The images were acquired with 60-fold magnification, a pixel size of 0.09 μm², a low flow rate, and high sensitivity. SpeedBead calibration reagents (400041, Luminex, USA) were used for daily calibration as internal beads and run concurrently for real-time velocity detection and autofocusing. The cell cultures were diluted with fresh medium 1–100-fold for measurement with an imaging flow cytometer. Approximately 10,000 cells (data points) were acquired and gated (Supplementary Fig. 1B) according to the fluorescence intensity and fluorescence aspect ratio intensity to exclude the internal beads and the cell culture debris with IDEAS software (v.6.2.183.0, Luminex, USA).

**Morphology analysis**. The cells (data points) used for the morphology analysis were further gated (Supplementary Fig. 1B) according to the sharpness quality of the cell images, i.e., the fluorescence gradient RMS (root mean square for image sharpness), as described previously[79]. Only the cell images in focus were used for the analysis. The relative lengths and widths of the cells were represented by the major and minor axis lengths, which were the longest and narrowest dimensions of the cell image, respectively. The cell shape was represented by the aspect ratio, which was the minor axis length divided by the major axis length and indicated the sphericity of the cell in the image. The relative cell size was represented by two features: area and volume. The relative cell area (*A*) was the total pixels of the cell image, and the relative cell volume (*V*) was calculated according to the relative length (*L*) and width (*W*) of the cells with the following formula (Eq. 1), as previously reported[80].

$$V = \frac{\pi}{6} \times L \times W^2 \qquad (1)$$

**Resource utilization assay**. The *E. coli* cells were inoculated from glycerol stocks into test tubes containing 3 mL of either 3.5 mM OAV-supplemented or 10.5 mM glucose-supplemented minimal medium and incubated in a bioshaker (MBR-022UP, Taitec, Japan) at 200 rpm and 37 °C as precultures. The precultures were subsequently transferred to 3 mL of fresh minimal medium that was supplied with 0.035, 0.07, 0.35, 0.7, or 3.5 mM OAVs or 0.105, 0.21, 1.05, 2.1, or 10.5 mM glucose, respectively, at a common dilution rate of 1000-fold. Every 10 test tubes of parallel cultures were applied for each concentration (a total of 10 concentrations). The changes in cell concentration, morphology, fluorescence, etc., were evaluated with an imaging flow cytometer at intervals of several hours until the cell culture reached the stationary phase. The utilization capacity was defined as the mean values (*N* > 5) of the maximal cell concentration (cells/mL), which were calculated according to the temporal measurements, divided by the concentrations (mM) of the supplied carbon source, i.e., OAVs or glucose. The steady population densities were measured, and the cell concentrations per mM carbon source were calculated as the population capacity. The results were summarized in Supplementary Data 6.

**Growth assays in glucose-supplemented minimal medium**. The *E. coli* cells were inoculated from glycerol stocks into test tubes containing 3 mL of 10.5 mM glucose-supplemented minimal medium and incubated in a bioshaker (MBR-022UP, Taitec, Japan) at 200 rpm and 37 °C as precultures. The precultures were diluted 1000-fold with fresh minimal medium supplied with 0.105, 0.21, 1.05, 2.1, or 10.5 mM glucose and subsequently loaded into flat-bottom 96-well microplates (Corning, USA) in six wells with locations varied per culture condition, as described previously[77]. The microplates were incubated in a plate reader (Synergy H1, BioTek, USA) with continuous orbital shaking at 282 cpm and 37 °C. Growth was monitored by measuring the absorbance at 600 nm, and readings were obtained at 30 min intervals for 20–30 h. The growth rate was calculated according to the changes in OD$_{600}$, as described previously[81].

**Scanning electron microscopy (SEM)**. The *E. coli* cells were fixed with 2.5% glutaraldehyde, followed by treatment with 1% OsO$_4$ for 1 h at 4 °C. Cells were then rinsed with PBS (phosphate-buffered saline) three times, dehydrated in a graded series (30%, 50%, 70%, 80%, 90%, 95%, and 100%) of ethanol, dried with a critical point dryer (Leica EM CPD 300, Leica Microsystems GmbH, Wetzlar, Germany), and coated with gold in a sputter coater (ACE600, Leica Microsystems). The prepared samples were observed using a scanning electron microscope (Hitachi S-4800, Japan) at an accelerating voltage of 3 kV.

**Genome mutation analysis**. The *E. coli* cells grown in glucose-supplemented minimal medium were harvested at the stationary phase for genome mutation analysis, as described previously[63]. Genome resequencing was performed by Sangon (Shanghai, China). Genomic DNA was extracted by a Magen Bacterial DNA KF Kit (Sangon, Shanghai, China), and gDNA libraries were constructed using the NEBNext Ultra DNA Library Prep Kit for Illumina (NEB, USA). Whole-genome resequencing was performed with the NovaSeq 6000 (Illumina, San Diego, CA) and MGISEQ-2000 platforms (MGI, Shenzhen, China) according to the manufacturer's instructions. Reads were mapped to the reference sequence (NCBI accession number NC_020518.1), and the genome mutations, i.e., SNPs and indels, were determined with the Genome Analysis Toolkit (GATK). RAW data sets were deposited at BioProject with accession number PRJNA693085 (SRR13487015-SRR13487022).

**Statistics and reproducibility**. All the biological experiments were performed repeatedly (*N* = 5–12). All the analyses were subjected to the statistic evaluation (*N* > 4) to draw the conclusion. The details were described in the corresponding sections of experiments and analyses. The data sets acquired from the repeated experiments and used for the statistic analyses are supplied as Supplementary Data for reference.

**Reporting Summary**. Further information on research design is available in the Nature Research Reporting Summary linked to this article.

## Data availability

Genome sequencing data are deposited at BioProject with accession number PRJNA693085. Source data used for the analyses and to generate the figures are available in Supplementary Data 1–6. Other data are available from the corresponding author upon reasonable request.

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

## Acknowledgements

The authors thank Yiwen Wang and Zhiwei Gong at the ECNU Multifunctional Platform for Innovation (004) Electron Microscopy Center for helping with the electron micrographs. The research was supported by the National Key R&D Program of China, Synthetic Biology Research (2019YFA0904500), and the MOE International Joint Laboratory of Trustworthy Software at ECNU and was partially supported by JSPS KAKENHI, Grant-in-Aid for Scientific Research (B) grant number 19H03215 (to BWY).

## Author contributions

B.W.Y. and T.Y. conceived the research; B.W.Y. designed the experiments; H.L. performed the experiments; H.L., M.K., H.A., and B.W.Y. analysed the data; M.K., Y.X., C.F., J.X., and L.K. provided the experimental and analytical tools; B.W.Y. drafted the paper and graphics; H.L., J.X., B.W.Y. and T.Y. revised the paper; and all the authors approved the paper.

## Competing interests

The authors declare no competing interests.
