## [Transparent Peer Review File · Communications Biology]

Reviewers' comments:

Reviewer #1 (Remarks to the Author):

Please see the attached review for comments on the manuscript.

Reviewer #2 (Remarks to the Author):

Lu et al. attempt to gain insight into the preferred shape of primitive organisms by "devolution" of *E. coli* in the presence of fatty acid vesicles, which are thought to have housed early protocellular systems. The work is interesting and seems to have been done carefully. The authors found a change in cell shape from rod to spherical and correlate this change to mutations in genes that code for the cell wall and protein synthesis. The work is insightful, even if the implications are not completely clear. The use of protocellular mimics as a food source to investigate "backwards" evolution can perhaps be a little misleading, as this work may have given insight into living things closer to LUCA but not necessarily into protocells.

"glucose-free but oleic acid vesicle (OAV)-rich conditions, which were considered as the laboratory primordial environments"

Probably some justification of this statement is warranted. I don't think I'd say that oleic acid vesicles are fantastic mimics of what was on the prebiotic Earth. In fact, I would suggest that the authors not make this argument. It would be more widely accepted if it were argued that fatty acid vesicles were likely present on the prebiotic Earth and so here oleate was used as an easy to use model prebiotic fatty acid.

Aren't spheres the thermodynamically preferred form? Therefore, doesn't that suggest that there is a selective advantage to the rod shape, since effort/energy must be expended to deviate from the thermodynamically favored form? I think this is in part why many were happy with spherical shapes in model protocells. It wasn't necessarily done on purpose. It's just what forms easily and thus was assumed to represent what would have formed prebiotically. For such reasons, I do suggest a little more caution in how the data are discussed by the authors. Also, could the authors speculate as to why the rod shape would be advantageous to later cells but not early cells?

Any idea as to how the different metabolic pathways exploited by living off of sugars vs. fats would either impact the shape or fitness of the cells?

Does "area" in the area-to-volume (A/V) ratio indicate the surface area?

Was there a reason why the *E. coli* strain MDS42ΔgalK::Ptet-gfp-kan was used? If so, it would be good to mention in the main text and not just the methods section.

Reviewer #3 (Remarks to the Author):

Exploring the shape of primitive life is an interesting subject. In this manuscript, authors performed the reverse evolution (or experimental devolution) using a modern bacterium *Escherichia coli* in the laboratory simulative primordial environment.

Some main issues I am concerned about are as follows:

- > First and foremost, I strongly suggest this manuscript should be substantially revised, and its logical flow and readability should be improved.
- > In this simulative study, the modern bacterium, *Escherichia coli*, was used as the object to explore the possible original shape of primitive life in the primordial environment. What I am concerned about is

why the modern bacterium, *Escherichia coli* was chosen as the target bacterium? Is there any pilot study? Maybe the authors should show some more information for the research object.

> More evidence should be provided to explain why glucose-free but oleic acid vesicle (OAV)-rich conditions was considered as the primordial environments, although authors stated that OAVs were proposed to be the major components as they were presumably available on early Earth. Except for carbon source, why other exogenous nutrients are not taken into consideration?

> Since a single colony could exhibit certain morphological features stimulated by environmental or physiological conditions, colony morphology has been regarded as a plastic trait. What I am concern about here is how long the shape of bacterium in this simulative study can maintain in the so-called laboratory primordial environment.

Reviewer #1

Article summary

*Here, I provide my review for the article titled: “Experimental devolution with a modern bacterium in a laboratory primordial environment.” The authors aim to answer the question: what was the shape of the first primitive cell? This is a very interesting question; all organisms are comprised of cells, and cell morphology – including shape and size – is varies to a large degree across the tree of life and underlies an organism’s reproductive success. To answer this question, the authors evolve the model organism *Escherichia coli* for a total of 500 generations in the presence of oil vesicles, those of which have properties that are believed to be representative of early earth conditions. At the end of the experiment, the “Devo” populations are subjected to a number of genomic and phenotypic analyses, including genetic sequencing, flow cytometry, microscopy, and growth assays. Major findings include evidence of adaptation to the OAV-rich environment, changes in cell shape and cell volume, and mutations within regulatory genes and those associated with cell wall.*

Response: Thank you for your careful reading and the interest to our study. Your valuable comments largely helped us to improve the manuscript. We hope the following point-to-point response and the revised manuscript meet your concerns.

Major comments:

Overall, I understood the experiments, and the figures – for the most part – are decent. The way the manuscript is framed, however, is not appropriate. The authors use very vague language throughout the manuscript to describe their results and exaggerate the meaning of their results (beginning from the article’s title itself). The manuscript needs to be reworked entirely and framed in a different way. To state all life evolved from a primitively spherical-shaped cell, given knowledge of physics and thermodynamics, is believable. To state that because all life evolved from a spherical shape using information for an organism that is removed from the first cell by billions of years (and using only one organism and without proper experimental controls) to draw those inferences is inappropriate.

Response: Thank you for the helpful comments. We are sorry for the poor writing. We do agree that the present study with *E. coli* might not represent the common principle regarding morphological determination of a primitive cell. The study just addressed the question of whether and how a rod-shaped bacterium change to sphere in laboratory by means of experimental evolution in the glucose-free and OAV-rich medium. A spherical shaped primitive cell was speculated for decades, probably just because of the stability in physics and thermodynamics of spheres. If it’s true, the morphological variation of modern cells would be attenuated, once they face to the primordial environment. The present study provides an experimental demonstration of the rapid change in morphology of *E. coli.*, from rod to spherical shape. In particular, the following discussion was newly added to the conclusion.

Lines 306-313:

*“The present experimental demonstration with *E. coli* might not represent the universality of morphological determinations of primitive cells; however, it shows some morphological devolution from rods to spheres by a modern bacterium surrounded by fatty acid vesicles. A spherical primitive cell was speculated for decades, probably because of the physical and thermodynamic stability of spheres. The present study provides a reasonable understanding of the growth benefits of spherical cells in primordial environments, and its findings strongly support this speculation, as did prebiotic studies.”*

Overall, to improve the clarity of the manuscript, the following changes were made in the revision: the title was changed; the figures and tables were modified and/or newly supplied; the Abstract and the Materials and Methods were modified; the Introduction and the Results

and Discussion were substantially revised with additional references. The changes made are highlighted in the manuscript.

Minor comments:

L1: “Experimental devolution” isn’t appropriate. You experimentally evolved populations of E. coli and spherical cells evolved. Bacteria have been on earth for billions of years. E. coli emerged about 120 million years ago. The experimental devolution title, and the language used throughout the manuscript, suggests you are making inferences for events that occurred in a far distant past using something that arrived much later in the biological and geological history of earth. I find those inferences unfitting to say the least. It would do this study much justice if you dial down on the grand big ideas you are reaching for and present the work as a test of adaptation on oleic acids. Here is a though: oleic acids have been shown to be extremely toxic to bacterial cells, yet it is believed that early earth conditions contained high concentrations of oleic acids in the environment, and that bacteria often consumed the substrate for growth and reproduction. If life were to persist under these conditions, bacteria needed to evolve strategies that overcome the toxicity. Here, we evolved Escherichia coli in the presence of OAVs to test the genomic and physiological changes associated with adaptation to an oleic acid rich environment. This is a much better framing than to say: we saw spherical cells evolve in our experiment; therefore, primitive cells were spherical. It is very circular logic. Reaching further back in the tree of life, i.e. working with archaeal organisms may provide a better answer. Another thing to consider are viruses. It is likely they were present before bacteria evolved. When did the rather complex capsule evolve in these organisms?

Response: Sorry for the improper term usage and thank you for your advice. The term of “experimental devolution” was deleted from the manuscript, and the corresponding paragraphs were rewritten.

Lines 70-97:

“To acquire such experimental evidence, experimental evolution with **modern living cells in the laboratory, induced by mimicking the resource regime of primordial environments, has been performed as a trial of devolution³². First, oleic acid vesicles (OAVs) were added as an easy-to-use model prebiotic nutrient to mimic the resources available in primordial environments.** The chemical composition of the primordial environment for **the growth** of primitive cells remained controversial^{33,34}. The first cellular lifeforms were supposed to consume the surrounding fatty acid vesicles for their growth when easily biodegradable small molecules were used up in the prebiotic soup^{35,36}. **As fatty acid vesicles were likely the major components present on the prebiotic Earth³⁷ and were adopted as the model membranes of protocells³⁸, oleic acid vesicles (OAVs) were easily adopted as the carbon resource. Second, E. coli is employed as the cell model, as it is the most representative bacterium of the stable rod shape^{39, 40}. The morphological change of E. coli from rod to filament is reported as a stress response to starvation⁴¹, which suggests that E. coli is able to change its shape when facing maladaptive resource utilization. Additionally, E. coli grows poorly on oleic acid, which is lethal to other model bacteria, e.g., Bacillus subtilis^{42, 43-45}. This result indicates that the evolutionary adaptation of E. coli to OAVs (as derivatives of oleic acid) is achievable within the experimental timescale. Finally, the experimental evolution of E. coli has employed OAVs as the only carbon source, replacing the commonly used glucose. The carbon source going into the central metabolism in modern cells is the representative nutrient used to trigger resource utilization stress and thereby trigger morphological changes in E. coli in experimental evolution studies; the easy-to-use glucose has been replaced by the unfavoured OAVs. That is, oleic acids, a rich source of nutrients on early Earth, are toxic to modern bacteria, but E. coli needed to consume them as a carbon source for growth, similar to what happens in adaptive evolution. In the present study, whether and how glucose-consuming E. coli could increase in**

growth fitness in a glucose-free but OAV-rich primordial soup and whether and how the morphology of rod-shaped *E. coli* changes during experimental evolution were investigated for the first time.”

In addition, thank you for the interesting idea of evolving the virus in a capsule. We have previously reported the experimental evolution regarding to the virus-like RNA self-replication (Ichihashi et al, 2013, Nat Comm; Bansho et al, 2016, PNAS) and the protocell-like fatty capsule (Ichii et al, 2013, Anal Chem; Tsuji et al, 2016, PNAS), which provided the genetic and/or prebiotic insights on the evolution of early life. The present study is trying to provide an alternative viewpoint on the properties of early life, i.e., the shape of early life. Although the morphological properties of early life remain largely argued and speculated, the latest study suggested that the protocell (LUCA) had an ovoid shape and the early bacterium (LBCA) was in rod shape (Baidouri et al., 10.1101/2020.08.20.260398). Taken together, we directly employed the bacterial cell to the present study, to avoid the biases and difficulties that we have experienced in addressing the genetic and prebiotic issues. Accordingly, the following sentence was newly added.

Lines 313-316:

“The ovoid-shaped protocell (LUCA) and the rod-shaped early bacterium (LBCA) were recently reported⁷³. Future studies combining synthetic cells and genetic materials are required to fully clarify the original cell shape.”

L54: Spheres are energetically stable due to surface tension. This is the reason why liquids form spheres (raindrops, oil, etc.) and planetary objects are roughly spheres. Moreover, spherical shapes place every dimension of the object at an equal distance from the center of the object. You might want to expand on this point and do a better job of communicating arguments for why a primitive cell may have been spherical. Also, you may contrast mechanisms then that lead to the rod-shape. What maintains the rod shape that exists across several microbial genera?

Response: Thank you for the intriguing thought. *Why a primitive cell may have been spherical* still maintains as an open question, although spherical shapes were employed in model protocells for decades in the field of the Origin of Life. It’s just what forms easily and thus was assumed to represent what would have formed prebiotically. Whether spheres are energetical or thermodynamically preferred by primitive cells remains unclear. Since a primitive cell is assumed to have no cell-wall, it might have formed the shape as similar as protoplast’s, which is roughly sphere. The rod-shaped bacteria generally have the machineries and/or pathways for synthesizing the building blocks of cell-wall, e.g., peptidoglycan. The mechanisms for cell-wall biosynthesis are supposed to be achieved during nature evolution and might be developed to *maintain the rod shape across several microbial genera*. The force to maintain the rod shape for genera might be the nutritional environment. The nutritional alteration could cause the changes in fatty acid synthesis in bacteria, and then affect the cell envelope capacity, which contributes to cell size and morphology (Vadia et al. 2017, Curr Biol). The rod shape might relate to the membrane capacity determined by the nutritional condition, which is rich in fatty acids on early Earth and sugars (i.e., glucose) on later Earth. Taking account of the concern, the following descriptions were newly added to the corresponding paragraphs.

Lines 17-21:

“What was the shape of primitive life? A spherical form was commonly presumed, which was often demonstrated using vesicles as synthetic cells in prebiotic studies but lacked experimental evidence in living cells. To investigate whether and how the shape of living cells changed, we exposed the rod-shaped bacterium *Escherichia coli*, a well-studied model cell, to a resource utilization regime mimicking a primordial environment.”

Lines 55-69:

“Considering the simplicity of the building blocks responsible for primitive cellular life, a spherical structure was assumed and has been employed in model protocells for decades in studies on the origin of life. This is why spherical-shaped compartments, e.g., vesicles and droplets²¹⁻²³, are commonly used to mimic protocells²⁴⁻²⁶. However, why a primitive cell may have been spherical and whether spheres were energetically or thermodynamically preferred by primitive cells are still open questions. Since a primitive cell is assumed to have had no cell wall, it might have taken on a spherical shape easily and simply, like the roughly spherical protoplast. In addition, the shapes of modern cells, e.g., bacteria, have been studied based on only genome homology²⁷. The only experimental demonstration was that in which L-form bacteria showed irregular morphologies due to deficiencies in the cell wall^{28,29}. Most modern bacteria have membrane synthesis machinery³⁰, which must have arisen during evolution to maintain their shape, e.g., the rod shape, in several microbial genera. Accordingly, the primitive cell without any evolved membrane and/or cell wall might have been spherical in shape³¹, but experimental evidence on the shape of primitive cells is needed.”

Lines 180-184:

“Alternatively, the changes in cell shape might be partially attributed to the changes in the plasma membrane capacity of *E. coli* caused by the carbon source changing from glucose to OAVs, because the changes in fatty acid synthesis due to such a nutritional alteration could influence the cell envelope capacity, thereby affecting cell size and morphology⁴⁸.”

Lines 298-306:

“Since morphology is regarded as a plastic trait, how long a spherical shape can be maintained in a primordial environment remains unclear. Theoretical prediction according to evolutionary dynamics (Fig. 1) showed that extended evolution would trigger increases in both the growth rate and the aspect ratio, independent of the carbon source (Fig. S12). However, the aspect ratio of the cells evolved in OAVs presented high variation among the six lineages, indicating the high plasticity of the spherical shape in the primordial environment. Although the fixed genome mutations indicated the stability of the sphericity, extended evolution in OAVs might lead to new mutations that disturb the morphological features and change the cell shape.”

L80-81: The way your text is worded, e.g. dilution rates and maintained in exponential phase, suggests these experiments were completed in a chemostat. Your figure suggests otherwise. Also, upon further examination of the figure and associated legend, it appears as though you transferred the cultures using different dilutions. You offer no explanation for why this was done. You would benefit from clarifying this information in the methods. As we know population size greatly determines the number of mutations that fix in a population via genetic drift. If you did not keep the dilution rate constant between your replicates you could be greatly biasing the number of mutations that fix in your independent replicates, as well as the number of generations that have surpassed in each. Please clarify.

Response: The serial transfer using different dilutions was to maintain the exponential growth phase, as previously reported (Kishimoto et al, PLoS Genet, 2015; Nishimura et al, 2017, mBio). It’s true that the population size affects the mutations fixation via genetic drift. Although the present study did not argue anything about genetics and/or molecular evolution, the inoculated population size of the daily transfer was in the range of 10^4 ~ 10^5 cells/mL, which was supposed to be large enough to avoid genetic drift in comparison to the final population size of 10^7 ~ 10^8 cells/mL. Accordingly, the revision was made in the Materials and Methods as follows.

Lines 371-373:

“Note that the initial cell concentration of the daily transfer was higher than 10^4 cells/mL, to avoid genetic drift.”

L88-89: Analyses of variance, or another appropriate statistical analysis on the distribution of growth rates is needed here.

Response: Thank you for the suggestion. As the experiments were newly performed, associated with additional analysis, the corresponding sentences were deleted in the revision. Instead, both the figure (please see Fig. 1) and the explanation were revised as follows.

Lines 109-123:

“The six lineages (L31, 32, 9~12), starting from a common originator (Ori), gradually increased in growth rate over the generations in the presence of OAVs (Fig. 1A, upper). The commonly improved fitness of the six populations (L#) demonstrated that glucose-consuming *E. coli* cells were able to utilize OAVs as their carbon source, which was likely an adaptation to the primordial-like environment rich in fatty acid vesicles. In comparison, the parallel experimental evolution in glucose (G31, 32, 9~12) presented higher growth rates than those seen in the OAV groups but similar dynamics as those in the OAV groups (Fig. 1A, bottom). Intriguingly, the magnitude of fitness improvement in the presence of OAVs was equivalent to that in glucose (Fig. 1B, upper), as the fold changes in growth rates of the evolved (Evo) and original (Ori) populations were not significant between the OAV and glucose groups ($p=0.1$). This demonstrated that the experimental evolution of a modern bacterium in the laboratory, induced by mimicking the carbon availability of a primordial environment, was applicable and comparable to the evolution in a regular environment with sugar as a carbon source.”

L103: It has been shown that changes in cell width supports better diffusion of substrates within the cell. Perhaps, stubby cells were selected for as they allowed for better diffusion of the rather large oleic acid molecules to diffuse within the cell?

Response: Yes, it's possible. It's always a puzzle for experimental evolution, whether the evolved population/mutant was selected from the initial population or newly born. As mentioned, the increased cell width might support better diffusion of substrates within the cell, whereas, the shorten cell length, observed in the present study (Fig. S8), probably mask such positive effect of diffusion within the cell. As OAVs could be trapped on the cell membrane due to the high hydrophobicity, the fatty cells might be selected. Taking account of the concern, the following sentences were added in the revision.

Lines 214-218:

“The raised aspect ratio was likely caused by the changes in cell length, as the mean length, i.e., the major axis of the cell, became shorter, whereas the width remained the same (Fig. S8). It might have been the changes in cell size attributed to the shortened cell length and not the diffusion effect mediated by the cell width that helped the cells utilize OAVs.”

L104-106: Not the correct figure to show differences between regressions. Why have you grouped the populations as they are? How much higher is one over the other?

Response: Sorry for the nonsense description. It's non-essential to group the populations. As the experiments and analyses were newly performed, the original discussion was deleted, and the corresponding paragraph was rewritten, as follows.

Lines 126-140:

“Whether the improved growth fitness was associated with morphological changes was analysed. The cell shape was evaluated by the aspect ratio, which represented the length ratio of the major to minor axes of the cell; that is, the closer to 1 the aspect ratio was, the more spherical the cell. A gradual increase in the mean aspect ratio of the cell population (L#) was commonly observed in the lineages evolved in OAVs (Fig. 1C, upper), unlike what occurred in the lineages evolved in glucose (G#) (Fig. 1C, bottom). Although the experimental evolution raised the aspect ratio independent of the carbon source (fold changes of Evo to Ori >1), the

change in cell shape that evolved in OAVs was significantly larger ($p=0.01$) than that in glucose (Fig. 1B, bottom). The cell morphology was further confirmed by single-cell imaging (Fig. 2, Fig. S2). The cells that evolved in glucose (G#) maintained a rod shape (Fig. 2B), similar to that of Ori cells (Fig. 2A). In contrast, those evolved in OAVs were all shorter and thicker than those evolved in Ori, and some of them were nearly spherical, regardless of whether grown in OAVs or glucose (Fig. 2C). These results demonstrate that rod-shaped *E. coli*, which generally maintains its shape while metabolizing glucose, became closer to spherical once adapted to OAVs.”

L128: Bacteria often produce larger cells at faster growth rates. Here, you evolved strains in OAV vesicles, where they increased in size, and then go on to assay these populations in media containing glucose. Where is the evolution in glucose only media control? A trade-off is expected to have occurred as the Devo strains have not been adapted to this environment. Therefore, growth rates should be slower and, as a consequence, cell size.

Response: Although we are wondering whether and why “bacteria produce larger cells at faster growth rates”, your idea of the additional experiment in glucose sounds nice to have a reasonable comparison. We additionally performed the experimental evolution in glucose and analyzed the data. In brief, the aspect ratio of the *E. coli* cells evolved in glucose increased slightly (Fig. 1), but still maintained in rod shape (Fig. 2). It indicated the change from rod to spherical shape was significantly occurred in the evolution in OAVs. The new experimental and analytical results were additionally provided as *highlighted* in the revised manuscript, the figures were revised (Figs. 1~3) associated with new supplementary tables (Tables S1~S5).

You also state that cells are much smaller and shorter in the OAV environment. How efficient is the machinery for utilizing this energy source? Cells under stress partition daughter cells that are roughly 1/4 the size of the parent cell (reductive division).

Also, the LTEE environment is not rich in glucose. In fact, there is much less carbon for LTEE populations to use within that experiment than provided here in this study.

Response: At the moment, we don’t know how efficient the cellular machinery utilizing OAVs as an energy source. The stress induced reduction in cell size (reductive division) might be the consequence of the energy saving in membrane synthesis but not carbon metabolism. As you mentioned, that the LTEE condition was not rich in glucose (carbon), however, it could lead to the increase in cell size (e.g., 10,000 generations in Lenski’s evolution). Although the mechanisms of either the increased sized in glucose (LTEE) or the decreased size in OAVs (the present study) are unclear, both findings commonly indicated the evolutionary selection in morphology. Taking account of the thought, the revision was made as follows.

Line 177-184:

“So far, whether and how OAVs impacted cell morphology through metabolism mediated by molecular machinery is unclear, as both the short-term evolution in OAVs and the long-term evolution in glucose^{46,47} cause changes in cell size. Alternatively, the changes in cell shape might be partially attributed to the changes in the plasma membrane capacity of *E. coli* caused by the carbon source changing from glucose to OAVs, because the changes in fatty acid synthesis due to such a nutritional alteration could influence the cell envelope capacity, thereby affecting cell size and morphology⁴⁸.”

L133: I expect for many features to be conserved among the Devo populations when assayed within the naïve environment they evolved in. Moreover, any changes associated with fixed mutations would also carry over between environments, until, given the fitness effect of those mutations, they are maintained in the “novel” environment or lost via purifying selection.

Later on, you show the populations have evolved mutations in the rod-maintenance genes. I suppose reverting those mutations back to wild-type, or complementation of those alleles with wild-type versions, would support a shift back toward rod-shaped cells. Moreover, if you had the glucose only evolution control, and a similar mutation evolved, you could have tested whether there is cross compatibility of specific mutations across environment directly.

Response: We do agree that the experimental verification (e.g., back mutation) of whether the mutated genes are responsible for the morphological changes would provide useful information for understanding the related molecular mechanism and gene function. In addition to the mutations that were known to participate the cell division and morphology (e.g., *mrdA/B* and *mreC*), largely varied mutations were detected in the six independent lineages. It indicated that there were the multiple genetic approaches for the morphological change from rod to spherical shape. Considering the complex interaction among the mutated genes for evolutionary adaptation (Suzuki et al, 2014, Nat Comm), it might be difficult to fully answer the questions raised here by genetic construction (e.g., back mutation). As the present study focused on the demonstration of the shape of early cell but not the genetic strategy for OAV adaptation please allow us to revise the discussion instead of the genetic construction for verifying the gene function. We hope the future work on the evolutionary mechanism could address the question of whether there is cross compatibility of specific mutations across environment directly. Taking account of these issues, the revision was made as follows.

Lines 287-297:

“The present study provides the first experimental evidence of rod-shaped bacterial cells changing to spherical shapes in a laboratory environment mimicking the resource regime of the primordial environment, in addition to the intensively reported prebiotic demonstrations of primitive cells. If the rod shape was attributed to the evolved cell wall and other cellular structures, the reverse change from rod to spherical form could be considered a breakdown of well-established cellular structures used by modern cells. The experimental evolution of the fitness-increasing changes in cell shape can be considered morphological devolution to a certain extent. The wide variety of mutations indicates the wide variety of genetic strategies for the evolution of primitive life. Future genetic reconstruction might provide reasonable clarification of the morphological change-associated adaptation to fatty acid vesicles.”

169-173: You state the “A/V ratio was disadvantageous for diffusion of small molecules,” and later on state the evolutionary change from rod shape to spherical shape “made the use of glucose unfavorable.” Did the authors test this explicitly? Sure, theory does indicate that cells with lower surface area to volume ratios will be diffusion limited, however, internal and external diffusion theory predicts two different things. You might want to either, a) test whether diffusion is slower in the Devo populations, or b) a better job of communicating to your readers that your interpretation is being grounded by theoretical predictions and not empirical tests. Also, see work done by Gallet et al. 2017 which discusses the benefits of evolving in either length or width and glucose diffusion.

Response: Thank you for the advice and the relevant reference. Our interpretation was being grounded by theoretical predictions but not empirical tests. At the moment, the diffusion rate could not be calculated, as the precise information of both internal and external cell mass and components was insufficient. Accordingly, the revision was made as follows.

Lines 222-225:

“The spheres presented a smaller A/V ratio than the rods⁵⁵, which was disadvantageous for the diffusion of small molecules, according to theoretical predictions^{56,57}. The rod shape with its higher A/V ratio was assumed to be advantageous for utilizing small molecules, e.g., glucose, but not for large molecules, e.g., oleic acids.”

180-181: OAV's are much bigger than glucose. Given your diffusion argument, I would expect for cells to favor larger A/V ratios in OAVs to enhance diffusion of the OAVs within the viscous cytoplasm of the cell. How are OAVs catabolized? Do OAVs need to be positioned in certain locations within the cell in order to be processed? Are they processed at the cellular membrane? It will help build better intuition for your readers if you describe OAV catabolism in the beginning of the manuscript, especially in light of knowing oleic acids inhibit cell growth.
<https://www.microbiologyresearch.org/docserver/fulltext/micro/91/2/mic-91-2-233.pdf?expires=1616014163&id=id&accname=sgid025916&checksum=A9E95341B7D06ADBBE3BECECEACDA9B1>

Response: Thank you for the helpful information. As suggested, we cited this paper and clearly stated in the introduction that oleic acids inhibit bacterial cell growth, with the recommended citation. The related discussion was added or revised as follows (some are partially repeated in response to other comments).

Lines 84-87:

“Additionally, *E. coli* grows poorly on oleic acid, which is lethal to other model bacteria, e.g., *Bacillus subtilis*^{42,43-45}. This result indicates that the evolutionary adaptation of *E. coli* to OAVs (as derivatives of oleic acid) is achievable within the experimental timescale.”

Lines 143-151:

“The changes in growth rate were highly correlated with those in aspect ratio and length, which were universally seen in all lineages evolved in OAVs (Fig. 3A, Tables S2), in contrast to the weak correlation in the lineages evolved in glucose (Fig. 3B, Table S3). The fitness increase was tightly associated with the changes in cell shape, which indicated that the consumption of OAVs required the cells to change from rods to spheres. Overall, the laboratory environment mimicking the carbon regime of a primordial environment provides experimental evidence of the sphericity of the cells surrounded by fatty acid vesicles, which supports the speculation of spherical primitive cells on prebiotic Earth.”

Lines 175-184:

“The changes in cell shape toward the spherical form rather than other morphological features were highly crucial for the glucose-consuming *E. coli* to use OAVs as the sole carbon source. So far, whether and how OAVs impacted cell morphology through metabolism mediated by molecular machinery is unclear, as both the short-term evolution in OAVs and the long-term evolution in glucose^{46,47} cause changes in cell size. Alternatively, the changes in cell shape might be partially attributed to the changes in the plasma membrane capacity of *E. coli* caused by the carbon source changing from glucose to OAVs, because the changes in fatty acid synthesis due to such a nutritional alteration could influence the cell envelope capacity, thereby affecting cell size and morphology⁴⁸.”

Lines 214-225:

“The raised aspect ratio was likely caused by the changes in cell length, as the mean length, i.e., the major axis of the cell, became shorter, whereas the width remained the same (Fig. S8). It might have been the changes in cell size attributed to the shortened cell length and not the diffusion effect mediated by the cell width that helped the cells utilize OAVs. In addition to the cell shape (i.e., aspect ratio), the surface area (A) and the volume (V) of the cells were analysed. The area-to-volume (A/V) ratios were significantly decreased with OAV supplementation (Fig. 6A, Fig. S9), indicating that the changes in cell shape from rods to spheres reduced the cell area to a greater degree than the cell volume. The spheres presented a smaller A/V ratio than the rods⁵⁵, which was disadvantageous for the diffusion of small molecules, according to theoretical predictions^{56,57}. The rod shape with its higher A/V ratio was assumed to be advantageous for utilizing small molecules, e.g., glucose, but not for large molecules, e.g., oleic acids.”

190: “Population density” is standard language. “Population capacity” is not. I might suggest changing it here, and throughout the manuscript, including the text in your figures and figure legends. After reading the manuscript methods, are you referring to the “carrying capacity” of the media at stationary phase? Please clarify.

Response: Sorry for the confusing term usage. “Population capacity” was replaced by “carrying capacity” in the following sentences.

Lines 189-193:

“The carbon utilization efficiency was quantitatively represented by the **carrying capacity**, that is, the maximal cell concentration (Fig. S5) per unit carbon source. The **carrying capacities** of the **Evos (evolved in OAVs)** showed rough increases with OAVs and decreases with glucose in comparison to those of the Ori (Fig. 5A).”

Lines 197:

“The trade-off in **carrying** capacity was linked to the trade-off in cellular protein abundance.”

Lines 241:

“It was highly intriguing that **an evolutionary** trade-off occurred in **carrying capacity** but not growth fitness.”

197: *It is a very rare event for a mutation to fix in the same position of a gene across multiple lineages. This means, then, that something must be said about a mutation that falls within the same gene multiple times. Your sentence, as it reads currently, negates the parallelism you observe (3/6 populations have a mutation in crp). You may want to calculate the probability of a mutation hitting a gene of said length within the genome. I bet there is a statistical significant association there.*

Response: We thank you for the helpful comment. The description was revised with additional citations as follows.

Line 260-267:

“**The alteration from rods to spheres of *E. coli* was associated with a wide variety of mutations without common mutations (Fig. 7), suggesting multiple genetic strategies for morphological changes. Intriguingly, the observed mutation was fixed within the same gene, cAMP-activated global transcriptional regulator *crp*^{62,63}, in three out of the six lineages. As mutations in *crp* have been reported in other evolution experiments with glucose⁶⁴⁻⁶⁶, transcriptional regulation by *crp* might be crucial for *E. coli* to use carbon sources efficiently. Morphological changes must have been associated with alterations in carbon metabolism to achieve balanced growth fitness.**”

You might also want to map the mutations on the protein structures and see whether they fall in any domains important to its activity, and use their locations to predict possible phenotypic effects.

Response: Thank you for the suggestion. The mutations were described in detail in the revised and/or newly supplied tables (Tables S4, S5).

*Also, more discussion could be had about *crp* with respect to its role in catabolite repression and your experimental findings. Could the mutations be allowing for simultaneous use of different carbon substrates, i.e., if you grew strains with the evolved *crp* alleles in both carbon sources studied here, would you see monophasic or biphasic growth?*

Response: Thank you for the advice. As the details of the gene functions and/or the molecular mechanisms are out of the scope of the present study, the discussion on the possible role of *crp* was added with the new citation as the same as the response to the previous comment.

Lines 260-267:

“The alteration from rods to spheres of *E. coli* was associated with a wide variety of mutations without common mutations (Fig. 7), suggesting multiple genetic strategies for morphological changes. Intriguingly, the observed mutation was fixed within the same gene, cAMP-activated global transcriptional regulator *crp*^{62,63}, in three out of the six lineages. As mutations in *crp* have been reported in other evolution experiments with glucose⁶⁴⁻⁶⁶, transcriptional regulation by *crp* might be crucial for *E. coli* to use carbon sources efficiently. Morphological changes must have been associated with alterations in carbon metabolism to achieve balanced growth fitness.”

Lastly, you have two different descriptions for crp in Table S2.

Response: Sorry for the careless mistake. It was fixed (Tables S4, S5).

L204: It is a far stretch to use words like “interrupt” when you don’t provide sufficient evidence of decreases in protein abundance for genes downstream of the regulators discussed. The direction of each of the downstream targets may be different (up vs. down regulated), and you don’t test that in this work.

Response: Thank you for the advice. “Interrupt” was replaced by the other term as follows.

Lines 273:

“A common genetic strategy was to target the genes participating in transcription and translation (Fig. 7, blue), ...”

Lines 282:

“It was reasonable that the cellular structure-related genes were targeted with high priority during the evolution ...”

Also, another peek at Table S2 shows many of the genes that you assign an “unknown” function for are indeed known. araC, for example is involved in arabinose fermentation. It looks like the mutations in those genes are in intergenic regions between the two genes listed. Please clarify.

Response: Sorry for the improper information. It was fixed (Tables S4, S5).

L214: Again, cell size and shape are common targets of selection. Several studies show cell size changes more generally with changes in growth rate, carbon source, competition with other organisms, oxygen tension and so much more. It is a very big leap to claim your observations of morphological changes in a mere 500 generation experiment provides the holy grail of evidence that the primitive cell was spherical.

Response: We are sorry for the misleading writing. The present study is NOT to demonstrate that the primitive cell was spherical but is try to provide the biological evidence, in addition to the plentiful prebiotic evidence, regarding the speculation on the sphericity of the primitive cell in the primordial environment in the field of the origin of life. At present, it’s hard to fully determine the primitive form, as it requires biological, physical and chemical analyses under the same experimental target and condition. Anyway, the experimental consequence of the spheres preferred by *E. coli* grown in OAV well supported the studies on the model protocells in spherical shape, which is just formed easily and thus is assumed to represent what would have formed prebiotically. Taking account of your concern, the manuscript was largely

rewritten. We hope the revision is acceptable. In particular, this paragraph was revised as follows.

Lines 287-316:

“The present study provides the first experimental evidence of rod-shaped bacterial cells changing to spherical shapes in a laboratory environment mimicking the resource regime of the primordial environment, in addition to the intensively reported prebiotic demonstrations of primitive cells. If the rod shape was attributed to the evolved cell wall and other cellular structures, the reverse change from rod to spherical form could be considered a breakdown of well-established cellular structures used by modern cells. The experimental evolution of the fitness-increasing changes in cell shape can be considered morphological devolution to a certain extent. The wide variety of mutations indicates the wide variety of genetic strategies for the evolution of primitive life. Future genetic reconstruction might provide reasonable clarification of the morphological change-associated adaptation to fatty acid vesicles.

Since morphology is regarded as a plastic trait, how long a spherical shape can be maintained in a primordial environment remains unclear. Theoretical prediction according to evolutionary dynamics (Fig. 1) showed that extended evolution would trigger increases in both the growth rate and the aspect ratio, independent of the carbon source (Fig. S12). However, the aspect ratio of the cells evolved in OAVs presented high variation among the six lineages, indicating the high plasticity of the spherical shape in the primordial environment. Although the fixed genome mutations indicated the stability of the sphericity, extended evolution in OAVs might lead to new mutations that disturb the morphological features and change the cell shape. The present experimental demonstration with *E. coli* might not represent the universality of morphological determinations of primitive cells; however, it shows some morphological devolution from rods to spheres by a modern bacterium surrounded by fatty acid vesicles. A spherical primitive cell was speculated for decades, probably because of the physical and thermodynamic stability of spheres. The present study provides a reasonable understanding of the growth benefits of spherical cells in primordial environments, and its findings strongly support this speculation, as did prebiotic studies. The ovoid-shaped protocell (LUCA) and the rod-shaped early bacterium (LBCA) were recently reported⁷³. Future studies combining synthetic cells and genetic materials are required to fully clarify the original cell shape.”

*L277: An appropriate control for your experimental setup is missing. That is, you do not evolve your populations in glucose only media. How does cell size and shape change under these conditions? Do the same mutations arise, or does selection act on other genes and associated traits? In the Lenski long-term evolution experiment, Grant et al. 2021 and Travisano & Lenski 1994, both show that cell size increases over the course of the experiment. Moreover, Grant et al. 2021, show cells increase in cell width during the first 10,000 generations of evolution, then reverting back to rod-shaped morphology by 50,000 generations. The mutations associated with these morphological changes are also discussed and include the rod-shape maintenance genes, *mrdA*, *mrdB*, *mreB*, *mreC*, and *mreD*. In other words, the above-mentioned authors report similar results, yet do not go as far to say that “spherical cells are the primitive form,” and that experiment has been ongoing for more than 150x generations longer. Without performing the control evolution experiment in glucose, I am forced to take the conclusions stated here with a grain of salt.*

Response: Thank you for the suggestion and discussion. To provide the reasonable comparison, the experimental evolution in the glucose medium was additionally performed, according to your comment. In brief, the *E. coli* cells remained in rod shape when evolved in the glucose condition, in comparison to those changed to spherical shape when evolved in the OAV condition. It indicated the morphological changes were specifically triggered by OAV. This was the experimental evidence of the *E. coli* cells evolved for ~500 generations, although it’s

unclear whether or how the cell shape changed again if the evolution experiment was extended for 10,000 or 50,000 generations, as what Lenski's group has done.

Again, we feel sorry for the misleading writing. The present study is NOT to conclude “spherical cells are the primitive form” but is to provide the biological demonstration of the prebiotic speculation on the sphericity of the primitive cell in the primordial environment. Thank you for the informative literatures, which provided us an interesting view on morphological changes of *E. coli*. That is, the experimental evolution in glucose (~10,000 generations) enlarged the cell width and size, whereas the experimental evolution in OAV (~500 generations) reduced the cell length and size, nevertheless, both led to the changes from rod to spherical shape. Of course, the experimental conditions (e.g., strains and media) were different between the present and Lenski's studies, the direct comparison of the studies was somehow improper. The mutated genes were partially overlapped between the two different studies, as you mentioned, indicated the morphology was one of the selective forces in evolution. According to the new results and your comments, the revision was sufficiently made in the manuscript and the figures and tables, as describe in response to the above comments.

L316: How is this equation derived? Please provide reference or proof. Also, does the equation provided here represent a spherical cell or a rod-shaped cell? See Ojkic et al. 2019 for well-defined equations for computing SA/V ratios from cell aspects. How do your SA/V measurements compare using eqn. 1 to those derived by Ojkic et al. 2019?

Response: Thank you for the literature information, which was newly cited in the revision. The cell volume (V) was represented by a sphere. The area (A) and the surface area (SA) was simulated and calculated by the imaging FCM, according to the manufacture's instruction. The equation used in the present study was exactly as same as that proposed by Ojkic et al in the literature, as described as follows.

Lines 403:

“The relative cell area (A) was the total pixels of the cell image, and the relative cell volume (V) was calculated according to the relative length (L) and width (W) of the cells with the following formula (Eq. 1) , as previously reported⁷⁸. ”

L319-320: I appreciate the authors standardizing the amount of carbon in the inoculum (63 mM for both oleic acid and glucose).

Response: The standard amount of carbon in the inoculum was confirmed.

L323-325: concentration of substrates should be in some appropriate order. I recommend putting them in ascending order. Currently, the values are not in a sensible ordination.

Response: Thank you for the advice. It was amended as follows.

Lines 411-412:

“... was supplied with 0.035, 0.07, 0.35, 0.7 or 3.5 mM OAVs or 0.105, 0.21, 1.05, 2.1 or 10.5 mM glucose, respectively, ...”

L339: Again, the concentration of substrates are not in any sensible ordination.

Response: It was fixed as follows.

Lines 427

“... supplied with 0.105, 0.21, 1.05, 2.1 or 10.5 mM glucose ...”

L357: Did you sequence clones from populations over the course of the evolution experiment, or at the final time point? Your study would benefit by analyzing clones along the line of descent

(100 generation intervals is recommended) using all methods presented in the manuscript (microscopy, flow cytometry, genomic and physiological assessments, etc.).

Response: Thank you for the suggestion. Besides the endpoint populations, the evolved populations at ~300 generations were sequenced. The results were summarized in Table S5, which was newly supplied in the revision.

Response to Reviewer #2

Lu et al. attempt to gain insight into the preferred shape of primitive organisms be "devolution" of E. coli in the presence of fatty acid vesicles, which are thought to have housed early protocellular systems. The work is interesting and seems to have been done carefully. The authors found a change in cell shape from rod to spherical and correlate this change to mutations in genes that code for the cell wall and protein synthesis. The work is insightful, even if the implications are not completely clear. The use of protocellular mimics as a food source to investigate "backwards" evolution can perhaps be a little misleading, as this work may give insight into living things closer to LUCA but not necessarily into protocells.

Response: It's our pleasure that you are interested in our study. Thank you for the insightful comments. We are sorry for the confusing term usage and unclear writing. To improve the clarity of our study and the manuscript, the following changes were made in the revision: the title was changed; the figures and tables were modified and/or newly supplied; the abstract and the materials and methods were revised; the introduction and the results and discussion were rewritten with additional references. The changes are highlighted in the manuscript. We hope the revised manuscript and the following point-to-point response will meet your concerns.

"glucose-free but oleic acid vesicle (OAV)-rich conditions, which were considered as the laboratory primordial environments"

Probably some justification of this statement is warranted. I don't think I'd say that oleic acid vesicles are fantastic mimics of what was on the prebiotic Earth. In fact, I would suggest that the authors not make this argument. It would be more widely accepted if it were argued that fatty acid vesicles were likely present on the prebiotic Earth and so here oleate was used as an easy to use model prebiotic fatty acid.

Response: Thank you for the helpful suggestion. We do agree that it will be more reasonable that "fatty acid vesicles were likely present on the prebiotic Earth and so here oleate was used as an easy to use model prebiotic fatty acid". Please allow us to adopt this statement in the revision. Besides, we would like to provide one more reason. OAV is known as not only the membrane material but also the energy resource, such as the carbon source for metabolism. As known, the carbon source (i.e., glucose) generally participates the central metabolism, which is the representative resource utilization in the modern living organisms. The present evolution experiment makes such a challenge to use OAV as the carbon source for *E. coli* growth, although OAV is unfavored by the modern bacteria. Taken together, the Abstract was revised as follows.

Lines 17-25:

*"A spherical form was commonly presumed, which was often demonstrated using vesicles as synthetic cells in prebiotic studies but lacked experimental evidence in living cells. To investigate whether and how the shape of living cells changed, we exposed the rod-shaped bacterium *Escherichia coli*, a well-studied model cell, to a resource utilization regime mimicking a primordial environment. Oleate was given as an easy-to-use model prebiotic nutrient, as fatty acid vesicles were likely present on the prebiotic Earth and might have been used as an energy resource. Six evolutionary lineages were generated under glucose-free but oleic acid vesicle (OAV)-rich conditions."*

Aren't spheres the thermodynamically preferred form? Therefore, doesn't that suggest that there is a selective advantage to the rod shape, since effort/energy must be expended to deviate from the thermodynamically favored form? I think this is in part why many were happy with spherical shapes in model protocells. It wasn't necessarily done on purpose. It's just what

forms easily and thus was assumed to represent what would have formed prebiotically. For such reasons, I do suggest a little more caution in how the data are discussed by the authors. Also, could the authors speculate as to why the rod shape would be advantageous to later cells but not early cells?

Response: Thank you for the intriguing questions of whether the spheres were thermodynamically preferred, why the rod shape would be advantageous to later cells but not early cells. As alternative questions, why a primitive cell may have been spherical, why the nature evolution led to the rod shape of the later cells, if the early cells had been spheres. These questions/issues are assumed to be highly related. Spheres are energetically stable due to surface tension. We do agree with you. It's unclear whether the spherical morphology is thermodynamically beneficial for *E. coli* grown in OAV, as well as, we are not sure whether the rod shape is advantageous for the modern cells. Both the rod shape resulted from the nature evolution and the spherical shape resulted from the experimental evolution are supposed to be largely dependent on the evolutionary environment, e.g., the surrounding nutrients available for utilization (the selective pressure as you commented). The following is our speculation in a biological insight.

To achieve the equivalent cellular protein concentration, which directly contributes to growth and metabolism, the spheres require fewer surface area than the rod shape does. It means that the spherical cell may save the material and energy for membrane synthesis, which is advantageous in either a resource-poor condition or an energy-costed resource-utilization environment. If the resource of fatty acids had been deficient on early Earth, the primitive cells might have preferred the spherical shape for resource-saving propagation. The utilization of OAVs might be the energy-costed metabolism for *E. coli* (as the devolved pathways in modern cells due to nature evolution), thus, the Devos must have favored the spherical shape to achieve energy-saving growth. If altering the cell shape has been the easiest way to regulate the cellular activity (metabolism) to achieve the efficient cell growth in response to the nutritional changes (both the richness and the types of the resource), it might be the reason why the modern cells are commonly evolved to have the molecular mechanisms (machineries) for size control, resultantly affecting the cell shape.

So far, it's hard to verify the hypothesis, as it requires biological, physical and chemical analyses under the same experimental target and condition. Anyway, the experimental consequence of the spheres preferred by *E. coli* grown in OAV well supported the studies on the model protocells in spherical shape, which is just formed easily and thus is assumed to represent what would have formed prebiotically, as you mentioned. Although both the primordial environment and the shape of the origin of life still remained speculated, the present study provided the first biological demonstration supporting the prebiotic speculation. Taking account of your concern, the additional sentences were added as follows.

Lines 30-34:

*“The morphological change of *Escherichia coli* for adapting to fatty acid availability supports the assumption of the primitive spherical form, which might have prevailed not only because of the thermostability of spheres but also because it was beneficial for energy-saving growth in primordial environments rich in fatty acid vesicles.”*

Lines 244-257:

*“Taken together, the findings suggest that the spherical form might save material and energy for membrane synthesis, which is advantageous in resource-poor conditions or when the cell undergoes energy-costing metabolism. If the resource of fatty acid vesicles had been deficient on early Earth, the primitive cells might have benefited from the spherical shape by saving resources for growth. The utilization of OAVs might be an energy-costing metabolic pathway for *E. coli* (as devolved pathways in modern cells due to natural evolution); thus, Evos must have favoured spherical shapes to achieve energy-saving growth. If morphological changes are*

the easiest way to regulate metabolism to achieve efficient growth in response to nutritional changes, it is reasonable that modern cells have evolved molecular machinery for size control, which also affects cell shape as a consequence⁴⁸. Although the nature of both the primitive cell and the primordial environment remains unknown, the present study provides a supportive demonstration of spherical protocells grown in a primordial environment rich in fatty acid vesicles.”

Any idea as to how the different metabolic pathways exploited by living off of sugars vs. fats would either impact the shape or fitness of the cells?

Response: It’s an interesting question of whether fats (OAV) impact cell morphology through metabolism. We currently have no experimental evidence on metabolic changes in the present case, nevertheless, the changes in fatty acid synthesis due to nutritional alteration could influence the cell envelope capacity, resultantly, affected cell size and morphology (Vadia et al. 2017, Curr Biol). The present finding of the changes in cell shape might be attributed to the changes in membrane capacity caused by the nutritional alteration from glucose to OAV. The following discussion was added in the revision.

Lines 177-184:

“So far, whether and how OAVs impacted cell morphology through metabolism mediated by molecular machinery is unclear, as both the short-term evolution in OAVs and the long-term evolution in glucose^{46,47} cause changes in cell size. Alternatively, the changes in cell shape might be partially attributed to the changes in the plasma membrane capacity of *E. coli* caused by the carbon source changing from glucose to OAVs, because the changes in fatty acid synthesis due to such a nutritional alteration could influence the cell envelope capacity, thereby affecting cell size and morphology⁴⁸.”

Does “area” in the area-to-volume (A/V) ratio indicate the surface area?

Response: Yes, the area indicated the surface area, which was evaluated by the imaging FCM. The explanation was newly added as follows.

Lines 218-219:

“In addition to the cell shape (i.e., aspect ratio), the surface area (A) and the volume (V) of the cells were analysed.”

*Was there a reason why the *E. coli* strain MDS42ΔgalK::Ptet-gfp-kan was used? If so, it would be good to mention in the main text and not just the methods section.*

Response: Sorry for the insufficient explanation. In the present study, the *E. coli* MDS42 was used as a simple cell model, because MDS42 has a smaller genome than the wild-type *E. coli* has and is IS-free, which benefit the genome resequencing analysis. In addition, using the genetically engineered MDS42 is because that it carries the chromosomally incorporated *gfp* (green fluorescent protein), which could be used as an indicator for cell detection and population analysis. The revision was made in accordance, as follows.

Lines 101-104:

“The laboratory *E. coli* strain MDS42ΔgalK::Ptet-gfp-kan was used as the cell model because the IS-free small genome of MDS42 was beneficial for precise genome resequencing analysis, and chromosomally incorporated *gfp* (green fluorescent protein) was practical as an indicator for cell detection and population analysis.”

Response to Reviewer #3

Exploring the shape of primitive life is an interesting subject. In this manuscript, authors performed the reverse evolution (or experimental devolution) using a modern bacterium Escherichia coli in the laboratory simulative primordial environment.

Response: Thank you for reading our paper and the valuable comments. We tried our best to address all the issues you raised. We hope the following point-to-point response and the revised manuscript will be satisfactory.

Some main issues I am concern about are as follows:

> First and foremost, I strongly suggest this manuscript should be substantially revised, and its logical flow and readability should be improved.

Response: We are sorry for the poor writing. The manuscript was largely rewritten to improve the clarity. The following changes were made in the revision: the title was changed; the figures and tables were modified and/or newly supplied; the abstract and the materials and methods were revised; the introduction and the results and discussion were rewritten with additional references. We hope the changes made (highlighted in the manuscript) improve the logic flow and readability and the revised manuscript meets your concerns.

> In this simulative study, the modern bacterium, Escherichia coli, was used as the object to explore the possible original shape of primitive life in the primordial environment. What I am concern about is why the modern bacterium, Escherichia coli was chosen as the target bacterium? Is there any pilot study? Maybe the authors should show some more information for the research object.

Response: Thank you for the questions, which helped us to notice the insufficient explanation of the experimental design. The reason why we used *E. coli* as the target is that *E. coli* is the most studied cell model in life science (e.g., genetics, adaption, evolution and metabolism, etc.). The new findings (results) based on the most representative bacterium might be common and well acceptable in the field. In the present study, the *E. coli* MDS42 was used as a simple cell model, because MDS42 has a smaller genome than the wild-type *E. coli* has and is IS-free, which benefit the precise genomic analysis. Additionally, the chromosomally incorporated *gfp* (green fluorescent protein) was employed as an indicator for cell detection and population analysis.

The other crucial reason is the availability of experimental demonstration. According to the knowledge on microbial fatty acid transport (López et al., 2021, Biotech Bioeng), *E. coli* is poorly but capable to grow on oleic acid, which indicates that the evolutionary adaption of *E. coli* to the glucose-free and OAV-rich medium is applicable. In comparison, oleic acid (OAV) is too toxic for some other model bacteria, e.g., *B. subtilis*, to achieve the evolutionary adaptation within the experimental timescale. Moreover, the bacterial cells could change their morphologies, e.g., filamentous, as a consequence of stress response (Wehrens et al., 2018, Curr Biol). When the rod-shaped *E. coli* cell is faced with the resource utilization stress, it might change its morphology, perhaps, to a spherical shape, for adaptation.

Taken together, we chose the *E. coli* MDS42 as the target bacterium for the present study. The revision was made in accordance, as follows.

Lines 80-87:

*“Second, *E. coli* is employed as the cell model, as it is the most representative bacterium of the stable rod shape^{39, 40}. The morphological change of *E. coli* from rod to filament is reported as a stress response to starvation⁴¹, which suggests that *E. coli* is able to change its shape when facing maladaptive resource utilization. Additionally, *E. coli* grows poorly on oleic acid, which*

is lethal to other model bacteria, e.g., *Bacillus subtilis*^{42, 43-45}. This result indicates that the evolutionary adaptation of *E. coli* to OAVs (as derivatives of oleic acid) is achievable within the experimental timescale.”

Lines 101-104:

“The laboratory *E. coli* strain *MDS42ΔgalK::Ptet-gfp-kan* was used as the cell model because the IS-free small genome of MDS42 was beneficial for precise genome resequencing analysis, and chromosomally incorporated *gfp* (green fluorescent protein) was practical as an indicator for cell detection and population analysis.”

> *More evidence should be provided to explain why glucose-free but oleic acid vesicle (OAV)-rich conditions was considered as the primordial environments, although authors stated that OAVs were proposed to be the major components as they were presumably available on early Earth. Except for carbon source, why other exogenous nutrients are not taken into consideration?*

Response: Thank you for your concern. We do agree that other exogenous nutrients must have existed in the primordial environment. Actually, it’s hard to say which chemical (nutrient) is the best candidate for the study. As known, our understanding of the primitive environment and the primitive cell are largely relied on the hypothesis and of limited experimental demonstration. How many kinds of the molecules (nutrients) surrounding the primitive cells remain largely unknown. In the present study, we tried to select a proper condition in the terms of experimental applicability and clear elucidation. The glucose-free and OAV-rich condition is one of the possibilities as primordial environment and is available for experimental demonstration.

There are two main reasons of choosing carbon source as the target nutrient. Firstly, as the carbon source, in particular, glucose, generally participates the central metabolism in the modern living organisms, it was chosen as the representative nutrient to trigger the resource utilization stress for morphological changes in the experimental evolution, in which the carbon source was replaced from the easy-use glucose to the unfavored OAV. Secondly, as oleic acid (OAV) is one of the major components available on early Earth and used as the common material for synthetic cells (protocells), it was considered as an appropriate resource used for the studies on the primitive cells. OAV is known as not only the material for membrane synthesis but also the energy resource, such as the carbon source for metabolism. The present study makes a challenge to use OAV as the carbon source for *E. coli* growth, although OAV is unfavored by the modern bacteria. The glucose-free and OAV-rich medium might be not a perfect and/or complete condition as a primordial environment, but is a practicable and representative condition mimicking the primordial environment in laboratory.

Taking account of your concern, the revision was made as follows.

Lines 72-80:

“First, oleic acid vesicles (OAVs) were added as an easy-to-use model prebiotic nutrient to mimic the resources available in primordial environments. The chemical composition of the primordial environment for the growth of primitive cells remained controversial^{33,34}. The first cellular lifeforms were supposed to consume the surrounding fatty acid vesicles for their growth when easily biodegradable small molecules were used up in the prebiotic soup^{35,36}. As fatty acid vesicles were likely the major components present on the prebiotic Earth³⁷ and were adopted as the model membranes of protocells³⁸, oleic acid vesicles (OAVs) were easily adopted as the carbon resource.”

Lines 87-94:

“Finally, the experimental evolution of *E. coli* has employed OAVs as the only carbon source, replacing the commonly used glucose. The carbon source going into the central metabolism in modern cells is the representative nutrient used to trigger resource utilization stress and thereby

trigger morphological changes in *E. coli* in experimental evolution studies; the easy-to-use glucose has been replaced by the unfavoured OAVs. That is, oleic acids, a rich source of nutrients on early Earth, are toxic to modern bacteria, but *E. coli* needed to consume them as a carbon source for growth, similar to what happens in adaptive evolution.”

> *Since a single colony could exhibit certain morphological features stimulated by environmental or physiological conditions, colony morphology has been regarded as a plastic trait. What I am concern about here is how long the shape of bacterium in this simulative study can maintain in the so-called laboratory primordial environment.*

Response: Thank you for the thoughtful comment. It’s an intriguing question of how long the changed cell shape (from rod to spherical) could maintain. According to the evolutionary dynamics of Aspect ratio (Fig. 1) and the theoretical regression (Fig. 2), the changed cell shape was supposed to be stable in the laboratory primordial environment (OAV-rich condition). Besides, the additional experiments showed that all Devos remained in spherical shape, even when the growth condition was changed from OAV-rich to glucose-rich media (Fig. 4). Considering the fact that the Devos fixed the genome mutations (Fig. 7, Table S4), which might contribute to the morphological changes, the spherical shape was assumed to be genetically stable. Of course, the extended evolution in the laboratory primordial environment might cause new mutations, which disturb the morphological features and change the cell shape. Taking account of the concern, the following discussion was added in the revision.

Line 298-316:

“Since morphology is regarded as a plastic trait, how long a spherical shape can be maintained in a primordial environment remains unclear. Theoretical prediction according to evolutionary dynamics (Fig. 1) showed that extended evolution would trigger increases in both the growth rate and the aspect ratio, independent of the carbon source (Fig. S12). However, the aspect ratio of the cells evolved in OAVs presented high variation among the six lineages, indicating the high plasticity of the spherical shape in the primordial environment. Although the fixed genome mutations indicated the stability of the sphericity, extended evolution in OAVs might lead to new mutations that disturb the morphological features and change the cell shape. The present experimental demonstration with *E. coli* might not represent the universality of morphological determinations of primitive cells; however, it shows some morphological devolution from rods to spheres by a modern bacterium surrounded by fatty acid vesicles. A spherical primitive cell was speculated for decades, probably because of the physical and thermodynamic stability of spheres. The present study provides a reasonable understanding of the growth benefits of spherical cells in primordial environments, and its findings strongly support this speculation, as did prebiotic studies. The ovoid-shaped protocell (LUCA) and the rod-shaped early bacterium (LBCA) were recently reported⁷³. Future studies combining synthetic cells and genetic materials are required to fully clarify the original cell shape.”

REVIEWERS' COMMENTS:

Reviewer #1 (Remarks to the Author):

Manuscript title is much improved.

L21: well-studied cell  well studied organism

L29: Consider... the mutations present in the genome revealed two distinct strategies of adaptation to OAV-rich conditions.

L30: The change in cell morphology. Morphological change is a little wordy.

L33: It is not clear what is meant by "energy-saving growth"

L41: Exploring the shape of primitive cells...

L62: Easily and simply?

L65: I am not sure I believe your claim about the only demonstration of cell morphology being studied in this context is with L-form bacteria. I would consider dialing back on this claim.

L76: Are supposed  were thought to.

L79: Here, is a rather weak transition from fatty acid vesicles present in the prebiotic environment and your rationale for using OAVs. Specifically, the "oleic acid vesicles were easily adopted as the carbon resource" is disjointed.

L88: "has employed?"

L89-95: This addition doesn't make sense.

L95: E. coli can use several other substrates for growth. Adding this adjective glucose-consuming makes it seem that this is the only carbon source it can use. Also, just because OAV was present in the early environment doesn't mean your experimental system truly replicates the primordial soup as I know it. I would stay away from that catch phrase. Suggestion for the paragraph: "In the present study, whether and how E. coli adapts in an OAV-rich environment and how cell morphology changes during experimental evolution under this condition were investigated for the first time." Even my correction needs to be wordsmith.

L101: You can't "make" the E. coli do anything. Experimental evolution of E. coli in an OAV-rich environment.

L105: We evolved six E. coli populations in either glucose- or OAV-supplemented media for approximately 50 generations.

L108: Again, does this means you grew strains in an environment akin to a chemostat?

L114: remove adjective glucose.

L126: Your subtitle needs to be changed here. I don't understand what is being said.

L134: How can change in cell shape be larger? Do you mean the change in cell size? Please clarify.

L152-153: I still would like to know how much energy a cell receives per mol of OAV. Cells often

become spherical under resource limiting conditions. See Harris and Theriot 2018.

L180: remove glucose adjective, here and throughout the manuscript. I won't comment on this further.

L191: reverse evolution implies you know with certainty what the primitive cell was. This is not true. Devolution and similar phrases are misconceiving. Please consider replacing them. Evolution is a forward moving process.

L198: I don't know what B-line fitting is and you offer no explanation of what it is in the manuscript.

L226: Your use of "which was" in this sentence makes it seem like you measured diffusion in this work, which you did not. Also, diffusion can be considered from both internal and external contexts. Make it explicitly clear which one of these you think is the most important in this system and why.

L233-239: It is common wisdom that cells become larger with nutrient upshifts. I suggest reviewing that literature.

Reviewer #2 (Remarks to the Author):

I find this work interesting. It's a study that I would have never thought to do myself, and does likely tell us some new things about evolution. The authors also seem to have put forth a good faith effort in answering the criticisms of the reviewers. My main concern now is readability, and that I feel that some aspects are a little too over interpreted. Backwards, laboratory evolution does tell us some things, but I'm not convinced that what is observed indicates what really transpired in the past. That doesn't mean that these experiments shouldn't be performed, but I do think that we should acknowledge the limitations of the approach. For example, I don't think it's universally accepted that lipids were used as a fuel source for early cells. I can see the logic for guessing that, that was the case, but I've never seen this investigated. Therefore, I think the arguments in the manuscript that makes such claims should probably be toned down.

I do like the argument about the changes in surface area to volume between rod and spherical shaped cells. But this would only be important if membrane transporters were lost during the backwards transition to a sphere, right? Were membrane protein transporters found to be mutated? It is also not clear to me why increased surface area to volume would be beneficial for small molecules but not for large molecules (as stated on page 8).

The very first sentence of the introduction "Exploring the shape of primitive life is crucial to understand the origin of life." does not seem substantiated. Why is this crucial to understand?

Generally, I think the interpretations should be more cautious. This study investigates what happens to *E. coli* when oleic acid is used as food source. We can make some guesses as to what that means for early life, which is fine, but we shouldn't get carried away.

Response to Reviewer #1:

Manuscript title is much improved.

Response: Thank you again for your valuable comments. The title was further revised according to the editor's suggestion, as follows: "*Inducing morphological change in Escherichia coli through primordial environmental mimic*".

L21: well-studied cell  well studied organism

Response: We deleted '*a well studied organism*' due to the limitation of Abstract (Line 21)

L29: Consider... the mutations present in the genome revealed two distinct strategies of adaptation to OAV-rich conditions.

Response: The sentence was revised accordingly, as follows: "*Highly differentiated mutations present in the genome revealed two distinct strategies of adaptation to AOV-rich conditions, i.e., either directly targeting the cell wall or not.*" (Lines 29-30)

L30: The change in cell morphology. Morphological change is a little wordy.

Response: Changed. (Line 31)

L33: It is not clear what is meant by "energy-saving growth"

Response: We deleted the last sentence due to the word limitation of abstract.

L41: Exploring the shape of primitive cells...

Response: Changed. (Line 38)

L62: Easily and simply?

Response: Changed to "easily". (Line 59)

L65: I am not sure I believe your claim about the only demonstration of cell morphology being studied in this context is with L-form bacteria. I would consider dialing back on this claim.

Response: We agree. We revised to "*As one of the experimental demonstrations, L-form bacteria showed irregular morphologies due to deficiencies in the cell wall*". (Lines 61-63)

L76: Are supposed  were thought to.

Response: Changed. (Line 74)

L79: Here, is a rather weak transition from fatty acid vesicles present in the prebiotic environment and your rationale for using OAVs. Specifically, the "oleic acid vesicles were easily adopted as the carbon resource" is disjointed.

Response: According to the suggestion, the sentence was revised as follows: "*Fatty acid vesicles were likely the major components present on the prebiotic Earth³⁷ and*

were adopted as the model membranes of protocells³⁸. As a representative model of fatty acid vesicles, oleic acid vesicles (OAVs) can be employed as one carbon resource for early life.” (Lines 76-79)

L88: “has employed?”

Response: Changed. (Line 86)

L89-95: This addition doesn't make sense.

Response: As suggested, it was deleted.

L95: *E. coli* can use several other substrates for growth. Adding this adjective glucose-consuming makes it seem that this is the only carbon source it can use. Also, just because OAV was present in the early environment doesn't mean your experimental system truly replicates the primordial soup as I know it. I would stay away from that catch phrase. Suggestion for the paragraph: “In the present study, whether and how *E. coli* adapts in an OAV-rich environment and how cell morphology changes during experimental evolution under this condition were investigated for the first time.” Even my correction needs to be wordsmith.

Response: We agree. The sentence was revised accordingly, as follows: “In the present study, whether and how *E. coli* adapts in an OAV-rich environment and how cell morphology changes during experimental evolution under this condition were investigated.” (Lines 87-89)

L101: You can't “make” the *E. coli* do anything. Experimental evolution of *E. coli* in an OAV-rich environment.

Response: The subtitle was revised accordingly, as follows: “Experimental evolution of *E. coli* in an OAV-rich environment” (Line 92)

L105: We evolved six *E. coli* populations in either glucose- or OAV-supplemented media for approximately 500 generations.

Response: The sentence was revised accordingly, as follows: “We evolved six *E. coli* populations in either glucose- or OAV-supplemented media for approximately 500 generations” (Lines 96-98)

L108: Again, does this means you grew strains in an environment akin to a chemostat?

Response: Yes, in the term of keeping exponential growth phase; but no, in term of keeping the cell concentration constantly.

L114: remove adjective glucose.

Response: Removed. (Line 104)

L126: Your subtitle needs to be changed here. I don't understand what is being said.

Response: It was changed as follows: “Fitness increase associated with changes in cell shape” (Line 117)

L134: How can change in cell shape be larger? Do you mean the change in cell size? Please clarify.

Response: The phrase was revised as follows: “*the magnitude of the change in aspect ratio*”. (Line 125)

L152-153: I still would like to know how much energy a cell receives per mol of OAV. Cells often become spherical under resource limiting conditions. See Harris and Theriot 2018.

Response: Yes, it’s an intriguing question. We hope the *E. coli* cells evolved in OAV would be the valuable materials for the studies on the relationship between the energy and the A/V ratio, as proposed in the Opinion by Harris and Theriots (reference no. 55 cited in the present study). As the energy calculation requires the precise measurement of cell size in μm^3 , which was unavailable in the present study, we hope the future work of physical and chemical analyses could address the question.

L180: remove glucose adjective, here and throughout the manuscript. I won’t comment on this further.

Response: It was removed.

L191: reverse evolution implies you know with certainty what the primitive cell was. This is not true. Devolution and similar phrases are misconceiving. Please consider replacing them. Evolution is a forward moving process.

Response: The confusing terms were deleted.

L198: I don’t know what B-line fitting is and you offer no explanation of what it is in the manuscript.

Response: Sorry for the improper name of the analytical method. It was revised as “cubic polynomial regression”. (Line 187)

L226: Your use of “which was” in this sentence makes it seems like you measured diffusion in this work, which you did not. Also, diffusion can be considered from both internal and external contexts. Make it explicitly clear which one of these you think is the most important in this system and why.

Response: Sorry for the confusing writing. The description was revised as follows: “As the spheres were thought to present a smaller A/V ratio than the rods⁵⁵, they might be disadvantageous for the diffusion of small molecules from external environment, according to theoretical predictions^{56,57}.” (Lines 215-217)

L233-239: It is common wisdom that cells become larger with nutrient upshifts. I suggest reviewing that literature.

Response: Thank you for the suggestion. The following sentence was added: “Although it was known that the cells became larger with nutrient upshift^{48,58}, it was unclear

whether the glucose-rich condition was nutrient upshift for the Evos, which were evolved in OAVs.” (Lines 226-228)

Response to Reviewer #2

I find this work interesting. It's a study that I would have never thought to do myself, and does likely tell us some new things about evolution. The authors also seem to have put forth a good faith effort in answering the criticisms of the reviewers. My main concern now is readability, and that I feel that some aspects are a little too over interpreted. Backwards, laboratory evolution does tell us some things, but I'm not convinced that what is observed indicates what really transpired in the past. That doesn't mean that these experiments shouldn't be performed, but I do think that we should acknowledge the limitations of the approach. For example, I don't think it's universally accepted that lipids were used as a fuel source for early cells. I can see the logic for guessing that, that was the case, but I've never seen this investigated. Therefore, I think the arguments in the manuscript that makes such claims should probably be toned down.

Response: Thank you for the valuable comments and the encouragement. The manuscript was revised accordingly, and the improper description, as mentioned, was deleted.

I do like the argument about the changes in surface area to volume between rod and spherical shaped cells. But this would only be important if membrane transporters were lost during the backwards transition to a sphere, right? Were membrane protein transporters found to be mutated? It is also not clear to me why increased surface area to volume would be beneficial for small molecules but not for large molecules (as stated on page 8).

Response: The mutations were occurred in the structural components of membrane but not the transporters, as described on page 9. It's also import that these genes are supposed to be absent in early life. Transport of resource molecules from external environment is discussed in the viewpoint of diffusion, because the early life might have utilized the resource in a simple manner. Small molecules are supposed to be dependent on diffusion, much more stringent than large molecules, which often require specific binding affinity by complex interactions. Thus, the increased A/V ratio benefits the small molecules more than the large ones.

The very first sentence of the introduction "Exploring the shape of primitive life is crucial to understand the origin of lie." does not seem substantiated. Why is this crucial to understand?

Response: A brief description was added as follows: "Exploring the shape of primitive cells is crucial to understand the origin of life, as it globally restricts physical and chemical features of a cell." (Lines 38-39). The detailed reasons have been provided later in the 2nd paragraph (Lines 50-61).

Generally, I think the interpretations should be more cautious. This study investigates what happens to E. coli when oleic acid is used as food source. We can make some guesses as to what that means for early life, which is fine, but we shouldn't get carried

away.

Response: Thank you again for your thoughtful comments helped us to improve the manuscript.